# CORE: Collaborative Optimization with Reinforcement Learning and Evolutionary Algorithm for Floorplanning

**Pengyi Li**[1]*, **Shixiong Kai**[2], **Jianye Hao**[1]†, **Ruizhe Zhong**[3], **Hongyao Tang**[1],
**Zhentao Tang**[2], **Mingxuan Yuan**[2], **Junchi Yan**[3]

[1] College of Intelligence and Computing, Tianjin University

[2] Huawei Noah's Ark Lab

[3] Dept. of Computer Science and Engineering & MoE Key Lab of AI, Shanghai Jiao Tong University

## Abstract

Floorplanning is the initial step in the physical design process of Electronic Design Automation (EDA), directly influencing subsequent placement, routing, and final power of the chip. However, the solution space in floorplanning is vast, and current algorithms often struggle to explore it sufficiently, making them prone to getting trapped in local optima. To achieve efficient floorplanning, we propose CORE, a general and effective solution optimization framework that synergizes Evolutionary Algorithms (EAs) and Reinforcement Learning (RL) for high-quality layout search and optimization. Specifically, we propose the Clustering-based Diversified Evolutionary Search that directly perturbs layouts and evolves them based on novelty and performance. Additionally, we model the floorplanning problem as a sequential decision problem with B*-Tree representation and employ RL for efficient learning. To efficiently coordinate EAs and RL, we propose the reinforcement-driven mechanism and evolution-guided mechanism. The former accelerates population evolution through RL, while the latter guides RL learning through EAs. The experimental results on the MCNC and GSRC benchmarks demonstrate that CORE outperforms other strong baselines in terms of wirelength and area utilization metrics, achieving a 12.9% improvement in wirelength. CORE represents the first evolutionary reinforcement learning (ERL) algorithm for floorplanning, surpassing existing RL-based methods. The code is available at https://github.com/yeshenpy/CORE.

## 1 Introduction

Electronic Design Automation (EDA) [1, 2] encompasses a series of steps from the initial phase of electronic system design to final manufacturing and testing. These steps include design, verification, synthesis, physical design (floorplanning, macro placement and routing), packaging, and manufacturing. Previous works have made significant progress in macro placement [3–11], whereas research on floorplanning receives less attention. **Unlike macro placement, floorplanning requires representations that support the construction of compact layouts, which makes macro placement algorithms ineffective for solving floorplanning problems. This is empirically validated in Section 4.2.** Floorplanning, as the first step of physical design, primarily involves determining the locations and shapes of functional modules, providing a foundation for subsequent placement, clock tree synthesis (CTS), and routing. Floorplanning is a crucial step in EDA, as it directly impacts

---

*Contact me at lipengyi@tju.edu.cn

†Corresponding author: Jianye Hao (jianye.hao@tju.edu.cn)

39th Conference on Neural Information Processing Systems (NeurIPS 2025).

subsequent placement, CTS, routing, and overall power consumption. However, the floorplanning problem is NP-hard [12], with a vast solution space, making it very challenging to construct a high-quality layout.

Methods for solving the floorplanning problem can generally be divided into three categories: SA-based algorithms, analytical methods, and reinforcement learning (RL) algorithms. SA-based methods [13, 14] model the problem based on specific floorplan representations, which encode the floorplan layout into a structure that is convenient for optimization , and rely on Simulated Annealing (SA) to search solutions [15, 16]. SA-based methods are reliable in small cases but easily converge to suboptimal solutions in large cases with vast search spaces. Analytical methods [17–21] require modeling the optimization objective as a differentiable function, which is then optimized through gradient descent. Analytical methods face challenges in constructing differentiable functions and are prone to getting stuck in suboptimal solutions due to the non-convexity of these functions [22]. For the third category, RL [23] is a class of learning algorithms that has achieved significant progress in various practical tasks. Using RL for floorplanning is an emerging research direction [24–28]. RL-based methods model the problem as a Markov Decision Process (MDP) [29] with specific floorplan representations, tackling the task by placing modules one by one and learning the solution through trial and error. RL can leverage fine-grained task feedback, which offers local guidance ability and better sample efficiency. However, RL heavily depends on the MDP modeling, which significantly affects solution quality. Additionally, the limited global exploration capability of RL can also lead to suboptimal solutions.

Previous work often suffers from insufficient exploration of the design space, leading to suboptimal results. We identify two main factors that impact the solution quality: (1) *Floorplan Representation*: Directly placing modules in continuous space is challenging. Floorplan Representation is used to create a more compact and favorable solution space. A proper floorplan representation should ease the solution-finding process with the purpose of avoiding overlap and meeting optimization objectives. The floorplan representation significantly impacts the feasibility and complexity of floorplan designs, further influences both the execution time and the quality of the results [30]. (2) *Limited Exploration Capability*: Previous methods often suffer from insufficient exploration capability and are prone to getting stuck in suboptimal solutions.

To address the above challenges, we propose a hybrid framework, named **C**ollaborative **O**ptimization with **R**L and **E**volutionary Algorithms (**CORE**). CORE employs B*-Tree representation [31], which offers a favorable solution space, linear transformation time, and compact placement. B*-Tree representation has been proven to outperform other representation methods. For efficient exploration, CORE integrates Evolutionary Algorithms (EAs) and RL. EAs [32] are a class of gradient-free heuristic optimization algorithms that mimic the process of biological evolution by maintaining a population. EAs improve the population through iterative variation. Compared to RL, EAs have better global optimization and search capabilities. CORE leverages the respective strengths of EAs and RL to achieve efficient layout optimization. It is worth noting that the integration of EAs and RL, known as Evolutionary Reinforcement Learning (ERL) [33–36], has become a vast research area. The most related prior works are those that use RL-assisted EAs to solve problems such as the Traveling Salesman Problem [36]. Yet, a collaborative framework for tackling more complex combinatorial optimization problems remains largely unexplored. In contrast, **CORE is the first attempt within the ERL domain to design a novel collaborative optimization framework for complex floorplanning tasks.**

CORE consists of three key processes: (1) Evolutionary Process, (2) Reinforcement Process, and (3) Collaboration Process. In the evolutionary process, we propose Clustering Diversified Evolutionary Search (CDES) based on B*-Tree representation. CDES maintains an initial population of layouts and clusters them into multiple subpopulations based on layout features. Each individual is ranked based on a weighted combination of its quality score and novelty score. Elite layouts are selected from each subpopulation and a new layout population is generated by perturbing these elites. Through the above iterations, CDES facilitates a more comprehensive exploration of the solution space. In the reinforcement process, we employ B*-Tree representation to model the floorplanning problem as a Markov Decision Process (MDP). We then adopt Proximal Policy Optimization (PPO) [37] to learn a policy network that makes step-by-step layout decisions. This process allows us to utilize finer-grained information for policy learning efficiently. To efficiently collaborate EAs and RL, we propose two collaboration mechanisms: (1) Reinforcement-driven Mechanism accelerates population evolution by injecting complete layout solutions obtained from RL into the population. (2) Evolution-

guided Mechanism maintains an archive of the best layouts discovered by EAs. The layouts that are better than RL's current performance are leveraged to guide the RL policy by imitation learning. Through these mechanisms, EAs and RL can achieve efficient collaboration. In the experiment, we evaluate our framework on two widely used benchmarks, MCNC and GSRC. Our experimental results demonstrate that CORE outperforms other strong baselines in terms of wirelength and area utilization metrics. Specifically, CORE achieves an average improvement of 12.9% in wirelength for each case compared to the best wirelengths achieved by other algorithms

We summarize the contributions as follows: (1) Firstly, we propose an EA-RL hybrid framework CORE and design collaboration mechanisms for efficient optimization. (2) Secondly, we propose the Clustering-based Diversified Evolutionary Search to maintain population diversity, thereby enhancing global optimization efficiency. (3) Thirdly, we model the floorplanning problem as a sequential decision problem using the B*-Tree representation. Then we employ PPO to learn a policy for floorplanning, aiming to enhance local guidance capabilities and improve optimization efficiency. To the best of our knowledge, we are the first to use RL with B*-Tree representation to solve the floorplanning problem. (4) Finally, experimental results demonstrate that CORE significantly outperforms other strong baselines, achieving a 12.9% improvement in wirelength.

## 2 Background

### 2.1 Floorplanning Problem

The core of the floorplanning problem lies in how to effectively arrange various functional modules to meet specific design objectives and constraints. The floorplanning problem involves the following three fundamental elements:

- Modules: Modules are the basic units in circuit design, denoted as $M = \{m_1, m_2, \ldots, m_{n_m}\}$. Each module $m_i$ is characterized by a rectangular shape with dimensions length $l_i$ and width $w_i$, centered at coordinates $(x_i, y_i)$.
- Ports: Ports are utilized for input and output signal transmission of current die, denoted as $P = \{p_1, p_2, \ldots, p_{n_p}\}$. Each port $p_i$ is located at a specific position $(x_i, y_i)$.
- Netlist: The netlist $E$ describes the connectivity within modules and ports. Each connection $e_i \in E$ represents a net that encompasses a set of modules and ports $\left\{m_1^{(e_i)}, m_2^{(e_i)}, \cdots, p_1^{(e_i)}, p_2^{(e_i)}, \cdots\right\}$.

A layout is formed after all the modules are placed. The optimization objectives are minimizing area, Half Perimeter Wire Length (HPWL), or other criteria. In these objectives, HPWL serves as a widely used indicator to assess the quality of a layout and is defined as follows:

$$\text{HPWL} = \sum_{e_i \in E} \left\{ \left( \max_{b \in e_i} \{x_b\} - \min_{b \in e_i} \{x_b\} \right) + \left( \max_{b \in e_i} \{y_b\} - \min_{b \in e_i} \{y_b\} \right) \right\} \tag{1}$$

### 2.2 B*-Tree Representation

B*-Tree [31] is an ordered binary tree that represents a compact placement. The process of converting a B*-Tree to a floorplan is illustrated in Figure 1. Each node $n_i$ in the B*-Tree represents a module $m_i$. We traverse the B*-Tree using Depth-First Search and determine the topological positions of each module according to the following rules: The root node of the B*-Tree corresponds to the bottom-left corner, i.e., $(0, 0)$. The left child $n_j$ of node $n_i$ represents module $m_j$ as the nearest right adjacent module to $m_i$, i.e.,

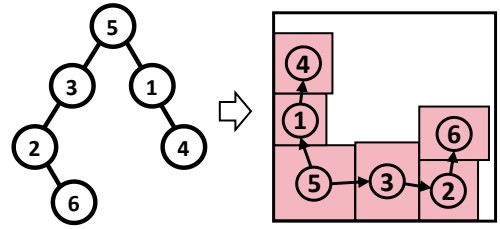

Figure 1: B*-Tree-based floorplan representation.

$x_j = x_i + w_i$. The right child $n_k$ of node $n_i$ represents the first module $m_k$ above $m_i$ with the same x-coordinate, i.e., $x_k = x_i$. To compute the y-coordinate, the B*-Tree utilizes a contour structure [38] — a doubly linked list of modules that records the contour line in the current compaction direction.

We choose the B*-Tree representation for three main reasons:

- **Efficient evaluation and update**. Since the MDP depends on the chosen representation, the computational cost of each state–action transition must be carefully considered. Unlike other representations, the B*-Tree avoids building constraint graphs or computing shortest paths, greatly reducing overhead. It also supports incremental evaluation, making it well-suited for sequential decision-making.

- **Compact search space**. A B*-tree encodes fewer permutations compared to most alternative representations, resulting in a more compact search space that facilitates efficient exploration and learning.

- **Modifiable open-source infrastructure**. Effective heuristics such as SA have already been developed for B*-Tree, and robust open-source implementations are readily available, making it easier to integrate and extend.

## 2.3 Evolutionary Algorithms

Evolutionary Algorithms (EA) [39, 40] are a class of black-box optimization methods. Below is a simplified optimization process: EA maintains a population of solutions $\mathbb{P} = \{s_1, s_2, \ldots, s_N\}$ in which evolution is iteratively performed. In each iteration, we first evaluate all solutions to get the fitness $\{f(s_1), f(s_2), ..., f(s_N)\}$ where $f(s_i)$ needs to be defined specifically according to the optimization objective. The solution with higher fitness is more likely to be selected as parents to produce the next generation with variation operators. The definition of the variation operator relies on expert experience tailored specifically to the problem. EAs iteratively optimize through the above process until a feasible solution is found.

## 2.4 Reinforcement Learning

Reinforcement Learning (RL) [23] is a category of learning algorithms primarily employed for solving sequential decision problems. RL involves modeling the problem as a Markov decision process [29], defined by a tuple $\langle \mathcal{S}, \mathcal{A}, \mathcal{P}, \mathcal{R}, \gamma, T \rangle$. At each step $t$, the agent uses a policy $\pi$ to select an action $a_t \sim \pi(\cdot|s_t) \in \mathcal{A}$ according to the state $s_t \in \mathcal{S}$ and the environment transits to the next state $s_{t+1}$ according to transition function $\mathcal{P}(\cdot|s_t, a_t)$ and the agent receives a reward $r_t = \mathcal{R}(s_t, a_t)$. The return is defined as the discounted cumulative reward, denoted by $R_t = \sum_{i=t}^{T} \gamma^{i-t} r_i$ where $\gamma \in [0, 1)$ is the discount factor and $T$ is the maximum episode horizon. The goal of RL is to learn an optimal policy $\pi^*$ that maximizes the expected return. PPO is a representative RL algorithm with convergence guarantees, which achieves significant success in various application scenarios. The key feature of PPO is the clipped objective function, which helps to prevent large policy updates that may destabilize training. This clipping mechanism ensures that the policy does not deviate too far from its previous version, thereby improving stability and convergence.

## 2.5 Evolutionary Reinforcement Learning

Evolutionary Reinforcement Learning (ERL) refers to a family of hybrid algorithms that integrate EAs with RL. It constitutes a broad research area encompassing multiple directions [36], including EA-assisted RL, RL-assisted EA, and synergistic optimization of EA and RL. In recent years, several ERL algorithms focus on integrating EAs and RL to achieve more efficient policy search. Representative methods such as ERL [41], CERL [42], PDERL [43], CEM-RL [44], ERL-Re$^2$ [45], VEB-RL [46], and EvoRainbow [47] show strong performance in various tasks involving games, locomotion, and manipulation. In addition, some works apply ERL-inspired ideas for reward function design, including Evo-Reward [48], Eureka [49], and R* [50]. Moreover, some works explore areas such as multi-agent systems [51, 52] and game testing [53]. In contrast, CORE focuses on complex combinatorial optimization problems with constraints. To the best of our knowledge, **CORE is the first ERL algorithm successfully applied to solving complex floorplanning problems.**

# 3 Collaborative Optimization with RL and EA

This section introduces our framework **C**ollaborative **O**ptimization with **R**L and **E**A (**CORE**). We begin by introducing the overall framework of CORE, followed by the details of its three key

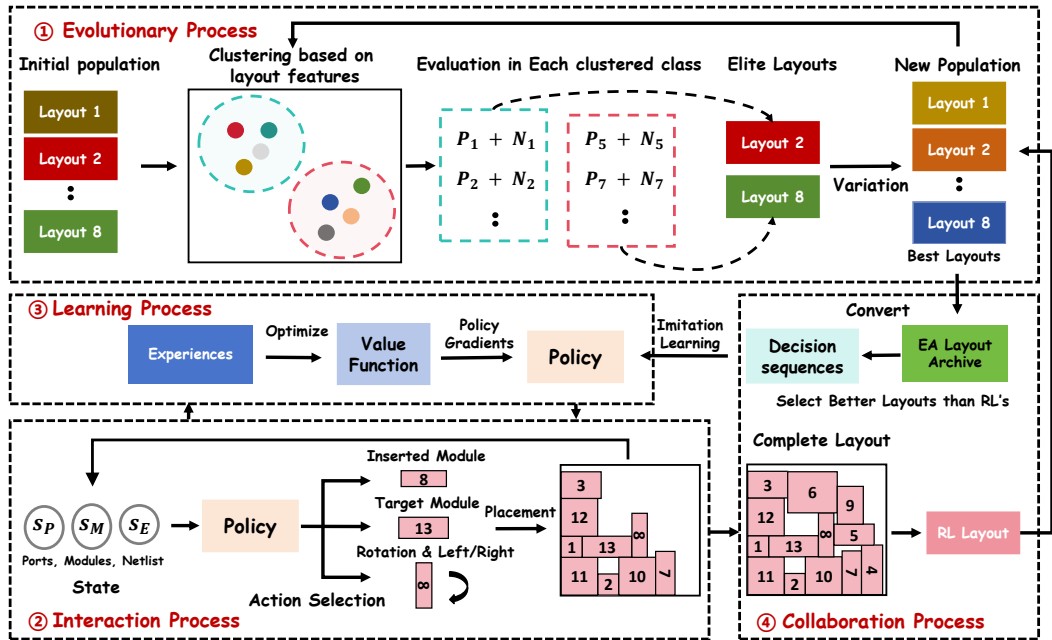

Figure 2: The framework illustration of CORE.

components: Clustering-based Diversified Evolutionary Search, B*-Tree-based RL, and EA-RL collaboration mechanisms.

## 3.1 Overall Framework of CORE

CORE is a novel ERL framework based on B*-Tree representation. The principal idea is to achieve efficient collaboration between EAs and RL for optimization, leveraging the strong search and global optimization capabilities of EAs along with the powerful learning abilities of RL. This synergy allows for efficient search and optimization within the vast solution space of floorplanning.

The framework illustration is shown in Figure 2, which comprises three parts: Evolutionary Process (Circle 1), Reinforcement Process (Circles 2 and 3), and Collaboration Process (Circle 4).

- The Evolutionary Process performs variation on the layouts within a population, aiming to search for layouts with higher quality scores and greater diversity.

- The Reinforcement Process requires the RL policy to interact with the environment and place modules step by step to construct a complete layout solution (Circle 2). Through these interactions, experiences are gathered to optimize the value function and policy (Circle 3).

- The collaboration process includes leveraging RL to accelerate population evolution and using EAs to guide RL policy learning.

In the following, we provide comprehensive introductions to the three key processes.

## 3.2 Clustering-based Diversified Evolutionary Search

Although EAs have strong search and global optimization capabilities due to the redundancy and randomness of their populations, relying solely on these properties can reduce diversity and make the population get stuck in suboptimal solutions. Improving population diversity can effectively solve this problem, as demonstrated by numerous Quality Diversity methods [54–57]. Motivated by this, we propose the Clustering-based Diversified Evolutionary Search (CDES). Specifically, as illustrated in Figure 2, CDES comprises five processes:

1. **Population Initialization**: Initialize the population $\mathbb{P} = \{s_1, \ldots, s_N\}$ with size $N$ where each individual $s_i$ represents a complete layout solution, all modules are planned.

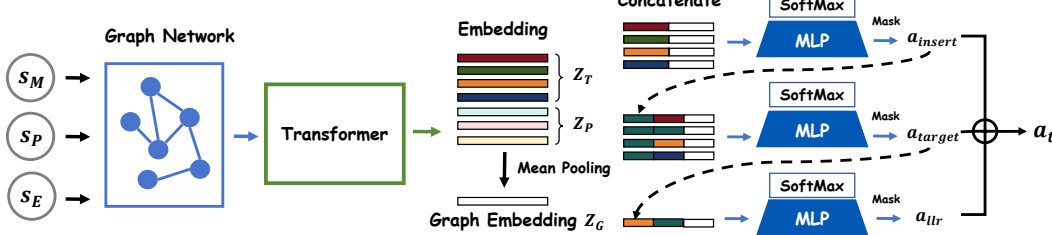

Figure 3: The policy architecture of PPO.

2. **Population Clustering**: We construct the layout features based on the positions of all modules, i.e., $\mathrm{F}(s_i) = \{x_{m_1}, y_{m_1}, \ldots, x_{m_{n_m}}, y_{m_{n_m}}\}$, and then utilize these features to cluster the population into $C$ clusters using K-Means [58].

3. **Subpopulation Evaluation**: Within each cluster, we evaluate the quality and novelty score of each layout. The novelty score $N(s_i)$ is defined as the negative average of the two highest cosine similarities between the current layout feature and the layout features of other individuals within the same cluster. Considering the top two similarities helps avoid situations where two layouts have the highest similarity to each other, making it challenging to rank them. Therefore, we consider the top two to enhance discrimination. The quality score $P(s_i)$ is constructed based on the optimization objectives, such as the negative of HPWL to minimize the wirelength.

4. **Elite Selection**: We combine the quality and novelty score to calculate the final fitness score, defined as follows:
$$f(s_i) = \alpha P(s_i) + (1 - \alpha) N(s_i), \tag{2}$$
where $\alpha$ is used to weight $P(s_i)$ and $N(s_i)$. A larger $f(s_i)$ indicates a better performance achieved by the layout and higher novelty simultaneously.

5. **New Population Construction**: We mutate the elite layouts within each cluster to generate offspring, forming a new population. The variation operation follows the principles of the B*-Tree-based SA [31] and includes three operators: ① Rotation Operator: randomly select a module and rotate it 90 degrees. ② Exchange Operator: randomly select two modules and swap them, ③ Deletion and Insertion Operator: remove a module that does not have both left and right children simultaneously, and insert the module into the B*-Tree as the left or right child of the target module.

Step 1 is executed only once at the beginning, while steps 2-4 are executed in a loop until the algorithm terminates. With CDES, we can maintain diverse layouts within the population, preserving population diversity and thus avoiding convergence to suboptimal solutions.

### 3.3 B*-Tree-based Reinforcement Learning

As heuristic algorithms, EAs rely on random exploration for layout optimization and often fail to leverage the experiences collected during the optimization process. These characteristics lead to inefficiency. In contrast, RL, as a learning algorithm, can utilize finer-grained information, approximate value functions, and optimize policies. Compared to EAs, RL offers more targeted optimization directions and stronger local optimization capabilities.

To fully exploit the strengths of RL, we model the floorplanning problem as an MDP based on the B*-Tree representation. The problem can be viewed as constructing a binary tree from scratch, wherein different modules are sequentially inserted into the tree. Specifically, we design the following five elements within the MDP $\langle \mathcal{S}, \mathcal{A}, \mathcal{P}, \mathcal{R}, T \rangle$:

- **States $\mathcal{S}$**: The state comprises information about all modules, ports, and netlist, i.e., $s = \{s_M, s_P, s_E\}$. Each module or port includes coordinates of the starting and ending points, length, width, whether it has been placed, whether it has a left child or right child in the tree, and the layout order (the order of adjustments in the B*-Tree). For ports, the length, width, presence of left/right child, and layout order are all set to 0. Netlist represents the connections between modules and ports.

- **Actions** $\mathcal{A}$: The action space comprises three types of actions: choosing the module to be inserted, selecting the target module, and determining whether the inserted module should be placed as the left or right child of the target module, along with the option to rotate the inserted module. Hence, the size of the action space is $4n_m{}^2$.
- **State Transition** $\mathcal{P}$: We adjust the layout according to the B*-Tree to avoid overlaps and obtain the adjusted layout.
- **Reward** $\mathcal{R}$: When we optimize the HPWL, we scale it with $10^{-4}$ and use the negative change in HPWL relative to the previous step as the reward, i.e., $r_t = -10^{-4}\Delta$HPWL, If none of the modules involved in a net $e_i$ are planned, the wirelength for that net is not calculated. When we optimize both HPWL and area utilization $U \in (0, 1]$, we define the area ratio as the area of the placed modules divided by the bounding box area, and use its change as the area reward. The final reward combines both terms: $r_t = -10^{-4}\Delta$HPWL $+ \Delta U$.
- **Maximum episode horizon** $\mathcal{T}$: When all modules are planned, the episode is done, so $\mathcal{T}$ equals $n_m$.

After modeling the MDP, we propose a policy architecture to process the complex state information. As depicted in Figure 3, we adapt a graph neural network [59] as the initial structure to process the current modules state $s_M$, the port states $s_P$, and netlist state $s_E$, Then we get the module embedding and port embedding by employing the Transformer [60] to process the output of the graph neural network. We use the mean pooling to obtain the current layout representation. Subsequently, we employ hierarchical conditional dependencies for action selection. Firstly, we concatenate representations of all modules with the current layout representation as input to MLP. Then, we select the inserted module by applying the action mask to remove actions corresponding to selecting modules on the panel. Secondly, we concatenate the inserted module representation with representations of all modules and the layout representation as input. Following the same procedure, we obtain the target module. Finally, we concatenate the inserted module representation, the target module representation, and the layout representation, and pass them through MLP and mask operator to obtain a four-dimensional input corresponding to the 4-dimensional action space, i.e., whether to rotate and whether to be the left or right child.

Based on the MDP modeling and the architecture design described above, we employ the PPO algorithm. The choice of PPO over other algorithms is motivated by several factors: 1) its guarantee of monotonic improvement and convergence, 2) its superior parallelization properties, and 3) the elimination of the need to maintain a state-action value function for the excessively large joint action space. For example, in a large case with 300 modules, the joint action space amounts to 360,000, making it challenging to approximate the values accurately. In addition to maintaining the policy as described above, PPO also requires the maintenance of a state value function. We use a separate representation processing module with the same architecture as the policy. The obtained layout representation is then passed through MLP to predict the value of the current state. We update the parameters of the policy network with the following loss:

$$\mathcal{L}_{\text{Clip}}(\theta) = \mathbb{E}\left[\min\left(\frac{\pi_\theta(a_t|s_t)}{\pi_{\theta_{old}}(a_t|s_t)}A^{\text{GAE}}(s_t, a_t), \quad \text{clip}\left(\frac{\pi_\theta(a_t|s_t)}{\pi_{\theta_{old}}(a_t|s_t)}, 1-\epsilon, 1+\epsilon\right)A^{\text{GAE}}(s_t, a_t)\right)\right], \tag{3}$$

where $\theta_{old}$ is the parameter of the policy before the update, $\epsilon$ is a hyperparameter used to constrain the update step size, and $A^{\text{GAE}}(s_t, a_t)$ is the advantage, which is calculated with Generalized Advantage Estimation (GAE) [61]. The state value network is optimized using the following loss:

$$\mathcal{L}_{\text{V}}(\phi) = \mathbb{E}\left[\left(V_\phi(s_t) - V_t^{\text{target}}\right)^2\right], \tag{4}$$

where $V_t^{\text{target}} = V_\phi(s_t) + A^{\text{GAE}}(s_t, a_t)$. To improve the exploration capability, we introduce an entropy loss to encourage the policy for more diverse exploration, defined as follows:

$$\mathcal{L}_{\text{E}}(\pi) = \mathbb{E}\left[\sum_a \pi(a|s_t)\log\pi(a|s_t)\right] \tag{5}$$

Based on the definition provided, we can learn a policy with PPO for the floorplanning problem. It is worth noting that the action space for the initial module differs from that of subsequent modules because there are no selectable target modules. Therefore, we randomly sample the initial module, while subsequent modules are selected through RL.

### 3.4 EA-RL Collaboration Mechanisms

In the previous two subsections, we explore solving the floorplanning problem using EAs and RL independently. However, these two processes are decoupled, and the performance of the final solution equals the best performance obtained by each method separately. To break through this performance bottleneck, we propose EA-RL collaboration mechanisms, which comprise 1) Reinforcement-Driven Mechanism and 2) Evolutionary-Guided Mechanism, as depicted in Figure 4. The Reinforcement-Driven Mechanism aims to leverage RL to accelerate the evolution of EAs. Specifically, we inject complete solutions constructed by RL into the population of EAs. Subsequently, based on the

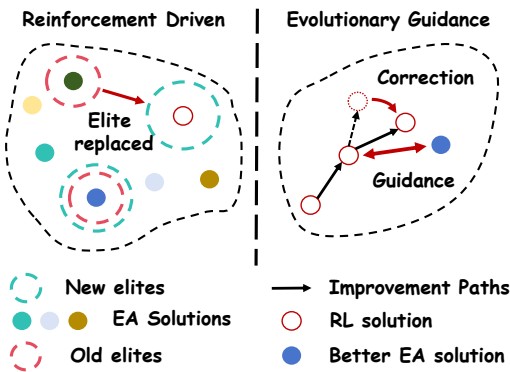

Figure 4: The EA-RL collaboration mechanisms.

defined fitness score, inferior layout solutions are removed to maintain a constant population size. If the solution discovered by RL achieves a higher score, it will replace other old elites in the population, thereby guiding population exploration. Conversely, if RL fails to provide better layout solutions, these layouts are promptly discarded without hindering the population's evolution.

The Evolutionary-Guided Mechanism is designed to utilize EAs to guide RL learning. For each module using different rotations as the root node, we store the best layout discovered by EAs into an archive $L_A$. Additionally, we also maintain the layouts constructed by RL with different root nodes. Then we compare the layouts in $L_A$ and the RL layout with the same root. If the EA layout is better, we convert it into a decision sequence (i.e., state-action pairs) based on the B*-Tree placement rule and store it in $D_{\text{EA}}$. Subsequently, we incorporate an Imitation Learning loss term into the PPO learning process, defined as follows:

$$\mathcal{L}_{\text{IL}} = \mathbb{E}_{s_t, a_t \sim D_{\text{EA}}} \left[ -\log \pi(a_t | s_t) \right]. \tag{6}$$

Through this approach, we enable RL to employ imitation learning to adjust the learning direction when the layouts it learns are inferior to those discovered by EAs. This allows RL to quickly grasp the superior layout and conduct more efficient local exploration around it. Thus, the final loss of PPO in CORE can be defined as:

$$\mathcal{L}_{\text{P}} = \mathcal{L}_{\text{Clip}} + w_{\text{Ent}} \mathcal{L}_{\text{E}} + w_{\text{Value}} \mathcal{L}_{\text{V}} + w_{\text{IL}} \mathcal{L}_{\text{IL}}, \tag{7}$$

where $w_{\text{Ent}}$, $w_{\text{Value}}$, and $w_{\text{IL}}$ are hyperparameters. The pseudocode is provided in Algorithm 1.

## 4 Experiments

### 4.1 Benchmarks & Baselines

Our experiments are conducted on two widely used floorplanning benchmarks: MCNC and GSRC. For the MCNC benchmark, we use the ami33 and ami49 designs. The GSRC benchmark includes six designs with module counts ranging from 10 to 300. The MCNC and GSRC benchmarks were designed based on the practical requirements of floorplanning. In industrial scenarios, floorplanning tasks typically involve fewer than 100 modules. In all cases, we reserve 10% of the area as whitespace and ensure a 1:1 aspect ratio for the boundary. We then project the ports to the new boundary. We consider two metrics for floorplanning: HPWL and Area Utilization (AU).

We compare with five competitive methods using different floorplan representations: Corblivar [14], SP-FP [13], B*-Tree SA [31], GoodFloorplan [27], and CBL-RL [28]. SP-FP, Corblivar, and B*-Tree SA are open-source. Both GoodFloorplan [27] and CBL-RL [28] are RL-based methods without open-source code. Based on the same setting, results from the literature are included in results tables. It is worth noting that they do not report area utilization, only HPWL. Therefore, we assume their area utilization to be 90%, which aligns with practical usage requirements. The convergence performance intuitively reflects the effectiveness of the algorithm. Previous methods all provide the best convergence performance. To enable a fair comparison, we train CORE and the

Table 1: Convergence Performance Comparison Based on HPWL and Area Optimization. CORE achieves shorter HPWL compared to other strong baselines while maintaining a comparable AU.

| Case | Corblivar HPWL | AU | SP-FP HPWL | AU | B*-Tree SA HPWL | AU | GoodFloorplan HPWL | AU | CBL-RL HPWL | AU | CORE HPWL | AU |
|---|---|---|---|---|---|---|---|---|---|---|---|---|
| n10 | 43082 | 0.897 | 36514 | **0.916** | 36044 | 0.859 | - | - | 41019 | 0.900 | **35170** | 0.911 |
| n30 | 124918 | 0.885 | 108835 | **0.913** | 111496 | 0.812 | - | - | 111793 | 0.900 | **99964** | 0.912 |
| n50 | 164044 | 0.902 | 137516 | 0.912 | 140716 | 0.851 | - | - | 162600 | 0.900 | **124890** | **0.948** |
| n100 | 271747 | 0.881 | 222559 | 0.911 | 218976 | 0.741 | 309320 | 0.900 | 337284 | 0.900 | **191349** | **0.915** |
| n200 | 535212 | 0.874 | 410438 | 0.912 | 388562 | 0.775 | 558330 | 0.900 | 351735 | 0.900 | **348956** | **0.917** |
| n300 | 764890 | 0.862 | 569820 | **0.910** | 626997 | 0.700 | 690760 | 0.900 | 476765 | 0.900 | **469242** | 0.901 |
| ami33 | 80298 | 0.830 | 64315 | 0.910 | 81629 | 0.869 | 87540 | 0.900 | 81767 | 0.900 | **52672** | **0.940** |
| ami49 | 1649560 | 0.822 | 930836 | 0.911 | 1180549 | 0.788 | 1067590 | 0.900 | 1375114 | 0.900 | **656068** | **0.918** |

Table 2: HPWL Comparison of CORE with macro placement methods.

| Method | n10 | n30 | n50 | n100 | n200 | n300 | ami33 | ami49 |
|---|---|---|---|---|---|---|---|---|
| MaskPlace | 54105 | 150706 | 201743 | 355918 | 666599 | 960487 | 126360 | 2410415 |
| ChipFormer | 54954 | 122035 | 165600 | 264180 | 500447 | 693501 | 88129 | 1377496 |
| CORE | **35170** | **99964** | **124890** | **191349** | **348956** | **469242** | **52672** | **656068** |

available open-source algorithm B*-Tree SA for the same number environment steps, running each algorithm five times and reporting the best convergence performance achieved. The experimental setting is consistent with prior literature. The average performance and error bar over five runs is provided in Appendix C (consistently outperforming the second-best baseline). We provide details on hyperparameter settings, network architecture, and training configurations in Appendix A. It is important to emphasize that floorplanning requires structural representations to achieve compact, gap-free placement, which makes macro placement algorithms inefficient for this task. We provide comparisons with the state-of-the-art macro placement algorithms in Appendix C.

## 4.2 Performance Evaluation

We compare CORE with other baselines on 8 cases. The experiment results in Table 1 show that CORE typically achieves comparable area utilization while maintaining a significantly shorter HPWL. CORE achieves an average improvement of 12.9% in wirelength compared to the best results obtained by other algorithms across all cases, with a 22.6% improvement on the n300 case and a 29.5% improvement on the ami49 case. This demonstrates that CORE has superior capabilities in layout search and optimization.

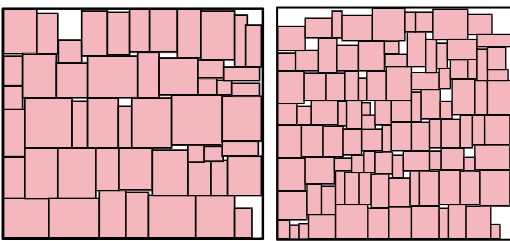

Figure 5: Visualization of the compact layout.

Besides, we observe that compared to B*-Tree SA, which uses the same floorplan representation, CORE demonstrates significant advantages in both HPWL and area utilization. Moreover, the performance gap increases with larger cases. This advantage primarily stems from CORE 's superior search and optimization capabilities, whereas B*-Tree SA is more prone to getting stuck in suboptimal solutions. When compared to the two RL methods, GoodFloorplan and CBL-RL, CORE also shows better performance metrics. The improvement stems from both the favorable solution space enabled by the B*-Tree representation and the complementary synergy between EA and RL in CORE. In addition, we provide the visualization results in Figure 5, demonstrating that CORE can achieve compact layout solutions. We also conduct experiments focusing solely on HPWL optimization in Appendix C. The results demonstrate that CORE consistently outperforms other baselines, achieving superior wirelength metrics, further validating its efficiency. Further details can be found in the appendix.

We also provide a comparison with macro placement methods. Unlike macro placement, the scale of floorplanning in real-world industry is typically under 100 blocks. Thus current floorplanning benchmarks are designed at this smaller scale. Floorplanning tasks generally rely on compact representations such as B*-Tree or CBL to minimize HPWL and area, ensuring gap-free placement.

In contrast, macro placement tasks often involve congestion constraints, requiring spacing between macros to allow for standard cell placement. This fundamental difference means that compact floorplanning representations are not suitable for macro placement tasks. Likewise, due to the lack of compact structural modeling, macro placement algorithms tend to perform poorly on floorplanning. To support our claim, we compare with two SOTA macro placement algorithms, MaskPlace [4] and ChipFormer [5]. The results are shown in Table 2. CORE, designed for floorplanning, significantly outperforms both MaskPlace and ChipFormer.

### 4.3 Analysis of CORE

In this section, we delve into each component of CORE to assess its significance. CORE comprises the evolutionary algorithm CDES, the RL algorithm PPO, and the collaboration mechanisms. We first compare the performance of CORE with CDES and PPO. **To intuitively reflect the algorithm's efficiency, we focus solely on minimizing HPWL in our comparisons.** Experimental results in Table 3 show that CORE achieves shorter HPWL compared to PPO and CDES. This highlights CORE's effective collaboration between RL and EAs, enabling it to surpass the performance of both methods. More hyperparameter analysis and ablation study are provided in Appendix C.

Table 3: Ablation study on CORE.

| Case | CDES | B*-Tree PPO | CORE |
|------|------|-------------|------|
| n10 | 35076 | 34604 | **34296** |
| n30 | 104045 | 99561 | **98713** |
| n50 | 126047 | 124683 | **123195** |
| n100 | 199488 | 189151 | **188376** |
| n200 | 382955 | 346382 | **342693** |
| n300 | 574579 | 468089 | **461219** |
| ami33 | 53773 | 49903 | **46849** |
| ami49 | 679473 | 642811 | **627620** |

In addition, we conduct a detailed analysis of CORE's collaboration mechanisms. We present the changes in the number of layouts maintained by the archive that outperform those constructed by PPO, along with the probability of layouts constructed by PPO being retained after injection into the population. The former indicates the extent to which evolutionary search can guide RL. Consistently low data volume suggests that EAs are unable to find layouts superior to RL, thereby failing to provide effective guidance. The latter reflects the extent of PPO's involvement in population evolution. If layouts constructed by PPO are immediately eliminated upon being added to

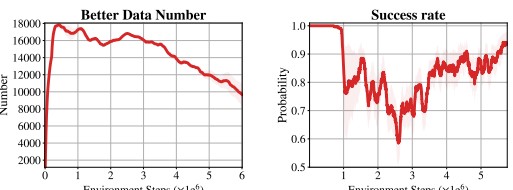

Figure 6: Analysis experiment on the collaboration mechanisms. The left side reflects the amount of data used for PPO imitation learning, while the right side reflects the success rate of RL-constructed layouts being injected into the EA population.

the population, it indicates that EA layouts are significantly superior. Consequently, PPO layouts cannot be injected into the population, demonstrating that PPO does not contribute to population evolution. In Figure 6, we present the results for the n100 case. We observe that the number of samples in the archive that outperform RL-constructed layouts consistently remains at a high level, indicating that EAs continuously provide effective guidance to RL. Additionally, the success rate of PPO layout injections into the population has consistently remained above 0.6, indicating that RL continues to influence the population evolution. The above two mechanisms complement each other, thus achieving better synergy. More experiments on runtime, CDES, and comparisons with ERL are provided in Appendix C.

### 5 Conclusion

This paper proposes an efficient hybrid framework, CORE, which combines EAs and RL to solve the floorplanning problem. CORE involves three main processes: the evolutionary process, the reinforcement process, and the collaboration process. Specifically, we introduce the clustering-based diversified evolutionary search to ensure robust global optimization capabilities. Additionally, we propose a complete reinforcement learning process with B*-Tree representation for floorplanning. Furthermore, we design effective collaboration mechanisms for EAs and RL. Our experimental results demonstrate that CORE outperforms strong baselines on two widely used floorplanning benchmarks.

## Acknowledgments

This work is supported by the National Natural Science Foundation of China (Grant Nos. 62422605, 624B2101, 92370132). We would like to thank all the anonymous reviewers for their valuable comments and constructive suggestions, which have greatly improved the quality of this paper.

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

---

**Algorithm 1:** Collaborative Optimization with RL and EA (CORE)

---

**1** **Initialize RL:** the PPO policy network $\pi_\theta$, the PPO value network $V_\phi$, the experience buffer $D$.

**2** **Initialize EA:** the population $\mathbb{P} = \{s_1, \ldots, s_N\}$ with size $N$, the $\alpha$ to combine quality scores and novelty scores, the number of clusters $C$, elite number $e$ in each cluster, The EA layout Archive $L_A$, EA experience buffer $D_{\text{EA}}$.

**3** **repeat**

**4**     # ① EA Process

**5**     **Population Clustering:** Perform K-Means clustering in the population based on the features $F(s_i)$ of each individual, resulting in the construction of $C$ clusters.

**6**     **Subpopulation Evaluation:** In each cluster, evaluate the quality score $P(s_i)$ and novelty score $N(s_i)$ of each layout

**7**     **Elite Selection:** Combine quality scores and novelty scores to obtain the final fitness: $f(s_i) = \alpha P(s_i) + (1 - \alpha)N(s_i)$, and then select the top $e$ elite layouts from each cluster based on the fitness.

**8**     **New Population Construction:** Mutate the elite layouts to create new layouts, replacing the non-elite layouts.

**9**     # ② Evolutionary-Guided Mechanism

**10**     Try to add the layouts within the population into the EA archive $L_A$ (Maintaining the best layouts for each module acting as the root, based on different rotations). Compare the layouts in $L_A$ with RL's layouts based on the same root node, and convert the EA better layouts into decision sequences, and then add the sequences into $D_{\text{EA}}$.

**11**     If $D_{\text{EA}}$ is not empty, calculate the imitation learning loss $\mathcal{L}_{\text{IL}}$ based on the experiences in $D_{\text{EA}}$.

**12**     # ③ RL Process:

**13**     **RL Interaction:** RL interacts with the environment to collect experiences, which are stored in $D$

**14**     **RL Optimization:** Optimize PPO using experiences from $D$ and the imitation learning loss $\mathcal{L}_{\text{IL}}$,

**15**     # ④ Reinforcement-Driven Mechanism: Add the complete layout constructed by RL to the population, calculate the score $f(s_i)$ for each individual, and keep the top $N$ individuals to ensure the population size remains unchanged.

**16** **until** *reaches maximum training steps*;

---

# A    Method Implementation Details

All experiments are carried out on NVIDIA GTX 2080 Ti GPU with Intel(R) Xeon(R) CPU E5-2680 v4 @ 2.40GHz.

## A.1    Baselines and Benchmark

We compare CORE with five baselines: Corblivar [14], SP-FP [13], B*-Tree SA [31], GoodFloorplan [27], and CBL-RL [28]. We utilize the open-source implementations of Corblivar[3], SP-FP[4], and B*-Tree SA[5]. For GoodFloorplan and CBL-RL, since the experimental setups are identical, we directly use the results from the original papers. Evaluation is conducted on two widely used benchmarks, MCNC[6] and GSRC[7]. **All algorithms are trained to convergence over 5 runs, and the best performances are reported.**

## A.2    Algorithm Pseudocode

The pseudocode for CORE is presented in Algorithm 1. The entire process of CORE contains three main parts: (1) Evolutionary Process, (2) Reinforcement Process, and (3) Collaboration Process. To improve clarity, we break down the Collaboration Process into the Evolutionary-Guided Mechanism and Reinforcement-Driven Mechanism.

Specifically, CORE begins with the Evolutionary Process. Here, population individuals are clustered to form subpopulations, evaluated within these subpopulations, elite individuals are selected, and a new population is constructed based on these elites. Subsequently, we initiate the Evolutionary-

---

[3]https://github.com/DfX-NYUAD/Corblivar

[4]http://staff.ustc.edu.cn/ songch/package.htm

[5]https://github.com/Ashley990921/B-Star-Tree

[6]http://vlsicad.eecs.umich.edu/BK/MCNCbench/

[7]http://vlsicad.eecs.umich.edu/BK/GSRCbench/

Table 4: Details of setting for the policy and value networks.

| Model | Selection |
| --- | --- |
| Activation Function | ReLU |
| GNN Type | SAGEConv |
| The layer number of the GNN | 2 |
| GNN input size | 10 |
| GNN hidden size | 64 |
| GNN output size | 64 |
| The layer number of the Transformer | 2 |
| The number of the Transformer attention head | 2 |
| Transformer input size | 64 |
| Transformer hidden size | 64 |
| Transformer output size | 64 |
| The layer number of the MLP for insert modules | 2 |
| MLP input size for insert modules | 128 |
| MLP hidden size for insert modules | 64 |
| MLP output size for insert modules | 1 |
| The layer number of the MLP for target modules | 2 |
| MLP input size for target modules | 192 |
| MLP hidden size for target modules | 64 |
| MLP output size for target modules | 1 |
| The layer number of the MLP for rotation and left/right | 2 |
| MLP input size for rotation and left/right | 192 |
| MLP hidden size for rotation and left/right | 64 |
| MLP output size for rotation and left/right | 4 |
| The layer number of the MLP for value prediction | 2 |
| MLP input size for value prediction | 64 |
| MLP hidden size for value prediction | 64 |
| MLP output size for value prediction | 1 |

Guided Mechanism, aiming to add population layouts into an archive $L_A$ based on their quality scores. The archive maintains the best layouts (If there are multiple optimization objectives, the Pareto frontier is maintained) for each module under different rotations as the root node. Then we compare the layouts in $L_A$ with RL's layouts and convert the layouts that are better (e.g., Pareto dominance) than RL's into decision sequences. The sequences are stored into a buffer $D_{EA}$. If $D_{EA}$ is not empty, calculate the RL imitation learning loss $\mathcal{L}_{IL}$ based on the experiences in $D_{EA}$.

Following this, we proceed to the Reinforcement Process. In this process, PPO interacts with the environment to collect experiences into $D$. We optimize PPO using collected experiences and the imitation learning loss $\mathcal{L}_{IL}$. Next, we activate the Reinforcement-Driven Mechanism to introduce fully constructed RL layouts into the population. Individuals in the population are evaluated based on their quality and diversity, yielding $f(s_i)$. The top $e$ layouts are retained to maintain the population size unchanged. The above process iterates until convergence.

## A.3 Network Architecture

This section provides a detailed description of our network architectures for PPO. In Figure 3, we present the specific process, and here we provide the detailed architectural design.

The PPO mainly consists of three parts:

- **Representation Module**: The graph neural network consists of two layers of SAGE convolutional layers with ReLU activation. Each module or port contains 10-dimension information, so the input dimension for SAGE is 10. The output dimension is 64. The Transformer employs PyTorch's integrated Transformer encoder layer with 2 layers, 2 attention heads, and a hidden size of 64. Therefore, each module or port representation is 64-dimension, and the graph representation, obtained by mean pooling across all modules and ports, is also 64-dimensional. The activation function is ReLU.

Table 5: Details of hyperparameters.

| Parameter | Value |
|-----------|-------|
| Optimizer | Adam |
| Learning rate | $2.5e - 4$ |
| Batch size | 128 |
| Gamma | 0.99 |
| GAE Gamma | 0.95 |
| Scale for HPWL | $1 \times 10^{-4}$ |
| Population size $N$ | 100 |
| Number of clusters $C$ | 4 |
| Elite size $e$ in each subpopulation | 2 |
| The weight to balance quality score and novelty score $\alpha$ in CDES | 0.8 |
| The weight to balance area utilization and HPWL in CDES | 0.1 for $U$ and 0.9 for HPWL |
| Clip ratio $\tau$ | 0.1 |
| The weight of entropy loss $w_{\text{Ent}}$ | 0.01 |
| The weight of value network loss $w_{\text{Value}}$ | 0.5 |
| The weight of imitation learning loss $w_{\text{IL}}$ | 0.1 |

Table 6: Algorithm Comparison Based on HPWL Optimization.

| Case | # Modules | # Nets | Corblivar | SP-FP | B*-Tree SA | CORE |
|------|-----------|--------|-----------|-------|------------|------|
| n10 | 10 | 118 | 43344 | 36204 | 34873 | **34296** |
| n30 | 30 | 349 | 131253 | 113498 | 106365 | **98713** |
| n50 | 50 | 485 | 171832 | 143920 | 130937 | **123195** |
| n100 | 100 | 885 | 266013 | 244956 | 223317 | **188376** |
| n200 | 200 | 1585 | 527828 | 464062 | 400560 | **342693** |
| n300 | 300 | 1632 | 753813 | 661199 | 621663 | **461219** |
| ami33 | 33 | 123 | 85116 | 57782 | 57815 | **46849** |
| ami49 | 49 | 408 | 1722410 | 913264 | 900802 | **627620** |

- **Decision Module**: The MLP input for deciding target modules is 128-dimensional, composed of 64 dimensions from module representations and 64 from graph representations. The MLP consists of two layers: input size 128, hidden layer size 64, and output size 1, determining the modules to be inserted. The inserted modules' representations are concatenated with the earlier 128-dimensional representation, yielding a 192-dimensional representation. This undergoes another two-layer MLP to select the inserted modules. The target module and inserted module representations, along with the graph representation, are concatenated and processed through a two-layer MLP with a hidden size of 64 and an output dimension of 4. This step determines actions such as whether to rotate and insert as the left or right child. All MLPs utilize ReLU as its activation function.

- **Value Network**: The value network starts with a separate representation module with the same structure. It takes modules, ports, and netlists as input to derive a graph representation. The graph representation then is fed into a two-layer MLP for value prediction. The MLP has an input dimension of 64, a hidden layer size of 64, and an output dimension of 1. ReLU serves as the activation function for the MLP.

## A.4 Hyperparameters

This section provides two aspects of hyperparameter selection: one for the evolutionary algorithm CDES and another for the RL algorithm PPO. Below, we present detailed hyperparameter choices.

For the CDES's hyperparameters, we define them as follows:

- Population size $N$ is set to 100.
- Number of clusters $C$ is 4.
- The number of elites in each subpopulation $e$ is set to 2.

- The balancing parameter $\alpha$, which balances quality and diversity, is set to 0.8.

- When optimizing solely for HPWL, its weight is 1.0.

- When optimizing for both HPWL and area utilization, we assign a weight of 0.1 to area utilization and 0.9 to HPWL.

For the selection of RL's hyperparameters, we define them as follows:

- Optimizer: Adam with a learning rate of $2.5 \times 10^{-4}$.

- Batch size: 128.

- Gamma: 0.99.

- GAE Gamma: 0.95.

- Clip ratio for PPO: 0.1

- The weight of entropy loss $w_{\text{Ent}}$: 0.01

- The weight of value network loss $w_{\text{Value}}$: 0.5

- The weight of imitation learning loss $w_{\text{IL}}$: 0.1

- When optimizing solely for HPWL, HPWL is multiplied by a scaling factor of $1 \times 10^{-4}$ as the reward at each step, i.e., $r_t = -10^{-4} * \text{HPWL}$.

- When optimizing for both HPWL and area utilization, the reward is computed by adding the area utilization $U$ and HPWL scaled by $1 \times 10^{-4}$, i.e., $r_t = -10^{-4}\text{HPWL} + U$.

Throughout the algorithmic process of CORE, RL interacts with the environment $I$ times per epoch, i.e., complete $I$ layouts. We set $I$ to 30, resulting in RL completing 30 full episodes of interaction with the environment. Following this, we inject $I$ complete layouts into the population. The number of evolutionary iterations equals the game length $T$ multiplied by $I$, divided by the population size $N$. This ensures that EAs and RL interact with the environment on as equal terms as possible from the perspective of environment steps. We train CORE on n200 and n300 for 20,000 epochs, while 10,000 epochs for other cases.

## B  Parallel Framework

We observe that due to the requirements of interaction and training, especially RL requires network forward passes for action selection. The algorithm is time-consuming. This is a common challenge faced by all RL-based algorithms.

To demonstrate this, we analyze the proportion of time cost for each component: RL interaction time accounts for approximately 65%, primarily due to the neural network processing inputs and selecting actions during each interaction, which is time-consuming. Training time accounts for 20.4%, and the evolutionary process accounts for around 15%. We can observe that the main source of time cost is the RL interaction.

To mitigate this, we develop a parallel framework, focusing on parallelizing interactions. This allows us to concurrently interact with multiple environments, thus significantly reducing runtime. Specifically, upon algorithm initiation, we initiate $L$ sub-processes. Subsequently, the main process sends the current network parameters of PPO to subprocesses, which initialize their networks based on these parameters and interact with the environment. Subprocesses collect complete experiences and return them to the main process. Once the main process gathers experiences from all subprocesses, it begins training the network parameters. With the aforementioned parallel interaction approach, we can greatly reduce the runtime. For example, on n30, the time consumption decreases from 11 hours to 3.78 hours, and on n50, it decreases from 23.6 hours to 8.3 hours. Additionally, we find that parallel sampling during the EA phase is possible. If the EA phase is also parallelized, the time for n30 will decrease from 3.78 hours to 2.54 hours, and for n50, it will decrease from 8.3 hours to 4.72 hours. but we have not implemented this version yet, leaving it for future work.

Table 7: Analysis of Population Size on HPWL.

| Pop size | n50 | n100 | ami49 |
|---|---|---|---|
| 50 | 126147 | 211774 | 711931 |
| 100 | 126047 | 199488 | **679473** |
| 200 | **124849** | **198223** | 680698 |

Table 8: Analysis of Cluster Number on HPWL.

| Cluster Number | n50 | n100 | n200 | ami49 |
|---|---|---|---|---|
| 1 | 132708 | 229737 | 417632 | 790673 |
| 2 | 127841 | 226644 | 403261 | 759635 |
| 4 | **126047** | 199488 | 382955 | **679473** |
| 10 | 144938 | **192398** | **357297** | 993230 |

# C  Additional Experiments

We also compare CORE with Corblivar, SP-FP, and B*-Tree SA in the setting where only HPWL is optimized. In this context, HPWL provides a more intuitive measure of the algorithm's search and optimization performance. We train all algorithms until convergence and report their convergence performance. The experimental results, shown in Table 6, show that CORE significantly outperforms the other algorithms, further highlighting the efficiency of CORE.

Here we present a hyperparameter sensitivity analysis. **It is worth noting that all hyperparameters remain fixed across all cases without any fine-tuning throughout the entire study.**

**Population Size**: We first analyze CDES independently (excluding the influence of RL), with the objective of minimizing HPWL. The results are shown in Table 7: We observe that larger population sizes generally lead to better performance.

**Cluster Number**: The results are shown in Table 8. For the number of clusters, a setting of 4 performs better on smaller-scale problems, while 10 is more effective for larger-scale instances.

**Novelty Weight**: The results are shown in Table 9. The novelty weight $\alpha = 0.8$ yields better performance.

**Elite size**: The results are shown in Table 10. n = 1 performs better on larger-scale tasks, whereas n=2 works better for smaller-scale cases.

**Imitation learning coefficient**: The results are shown in Table 10. For the imitation learning coefficient, we observe that the algorithm performs best when the coefficient is set to 0.1.

**It is worth noting that we did not fine-tune these parameters—across all tasks, we consistently used population size 100, 4 clusters, elite size 2, $\alpha$ = 0.8 and w=0.1.**

In the main text, we report the best results following prior works. Below, we present the mean and standard deviation across five runs, as well as a performance comparison with the second-best floorplanning algorithm. The results are shown in Table 12. We observe that CORE consistently outperforms the best performance of other baselines.

**Runtime comparison between CORE and other methods.** CORE introduces RL, which results in additional time overhead compared to heuristic methods. Thus we design a parallel architecture to improve efficiency. Unfortunately, existing learning-based methods for floorplanning have not released open-source implementations. Therefore, we compare the runtime and achieved HPWL with the available baselines in Table 13. CORE achieves better HPWL compared to baselines, and requires more time than EA. However, it outperforms other learning-based method MaskPlace.

**Ablation study on CORE.** We present the results of our ablation study in Table 14, which clearly demonstrate that the integration of both mechanisms leads to improved performance.

**Comparison between CORE and ERL.** We additionally implement a parameter-centric ERL framework. It is important to note that such frameworks typically rely on off-policy algorithms, such

Table 9: Analysis of $\alpha$ on HPWL.

| $\alpha$ | n50 | n100 | n200 | ami49 |
|---|---|---|---|---|
| 0.0 | 138954 | 234500 | 408446 | 837418 |
| 0.2 | 138592 | 231034 | 411872 | 805013 |
| 0.5 | 136496 | 225504 | 418917 | 766239 |
| 0.8 | **126047** | **199488** | **382955** | **679473** |
| 1.0 | 191069 | 341052 | 835619 | 1729777 |

Table 10: Analysis of elite size $e$ on HPWL.

| Elite size $e$ | n50 | n100 | n200 | ami49 |
|---|---|---|---|---|
| 1 | 135760 | **191708** | **359261** | 824228 |
| 2 | **126047** | 199488 | 382955 | **679473** |
| 5 | 130018 | 225533 | 410475 | 731270 |

as TD3 or SAC, which face inherent challenges when applied to discrete action spaces. To ensure a fair and consistent comparison, we adapt the baseline methods to use B* tree-based PPO, which is better suited for discrete structural representations. The corresponding experimental results are presented in Table 15: We observe that CORE consistently outperforms the ERL-based methods across all cases.

**Comparison between CDES and EA.** We compare our proposed CDES with a vanilla EA that does not incorporate the clustering method. The results in Table 16 show that CDES achieves significantly better optimization performance than the vanilla EA.

**Comparison between B* Tree and other representation.** We evaluate the effectiveness of the B*-Tree modeling approach against alternative structure Sequence pair. The results in Table 17 indicate that the B*-Tree model consistently yields superior performance.

**Ablation Study on the Transformer Module.** It is important to note that the network input includes a netlist, whose topological structure naturally lends itself to graph neural networks (GNNs). Therefore, the use of a GNN is essential. We introduce transformer layers primarily for better representation. The ablation study is shown in Table 18. We observe that the Transformer module contributes to better model performance.

Table 11: Analysis of $w_{IL}$ on HPWL.

| $w_{IL}$ | n50 | n100 | ami33 | ami49 |
|---|---|---|---|---|
| 1.0 | 124319 | 189956 | 48437 | 634400 |
| 0.1 | **123195** | **188376** | **46849** | **627620** |
| 0.01 | 124054 | 189103 | 47619 | 631599 |
| 0.0 | 124172 | 189009 | 48403 | 635198 |

Table 12: Comparison Between CORE and the Best Baseline on HPWL (mean ± std).

| Method | n10 | n30 | n50 | n100 | n200 | n300 | ami33 |
|---|---|---|---|---|---|---|---|
| Best baseline | 36044 | 108835 | 137516 | 218976 | 351735 | 476765 | 64315 |
| CORE | $35290 \pm 294$ | $100733 \pm 1113$ | $125132 \pm 473$ | $191610 \pm 840$ | $348416 \pm 1140$ | $470296 \pm 701$ | $52930 \pm 753$ |

Table 13: Runtime and HPWL Comparison between CORE and baselines.

| Case | EA | Maskplace | CORE |
|---|---|---|---|
| n30 | 1.65 \| 104045 | 4.15 \| 150706 | 3.78 \| 99964 |
| n50 | 3.54 \| 126047 | 6.37 \| 201743 | 8.3 \| 124890 |

Table 14: Ablation study of CORE on different cases.

| Method | n30 | n50 | ami33 | ami49 |
|---|---|---|---|---|
| CORE w/ Injection | 98888 | 123938 | 49061 | 641670 |
| CORE w/ IL | 99302 | 124172 | 48403 | 635198 |
| CORE w/o both | 99561 | 124683 | 49903 | 642811 |
| **CORE** | **98713** | **123195** | **46849** | **627620** |

Table 15: HPWL Comparison between CORE and ERL.

| Method | n10 | n30 | n50 | n100 |
|---|---|---|---|---|
| ERL | 34645 | 99911 | 124418 | 190343 |
| CORE | 34296 | 98713 | 123195 | 188376 |

Table 16: Comparison between CORE variants with EA and CDES.

| Method | n30 | n50 | ami33 | ami49 |
|---|---|---|---|---|
| CORE w/ EA | 101252 | 127929 | 52604 | 698285 |
| CORE w/ CDES | 98713 | 123195 | 46849 | 627620 |

Table 17: HPWL Comparison between Sequence Pair and B* Tree Representations ($\times 10^3$).

| ($\times 10^3$) | n100 | n200 | n300 | ami33 |
|---|---|---|---|---|
| Sequence pair | 209.8 | 404.5 | 556.9 | 59.0 |
| B* tree | 191.3 | 349.0 | 369.2 | 52.7 |

Table 18: Ablation Study on the Transformer Module.

| Method | n30 | n50 | ami33 | ami49 |
|---|---|---|---|---|
| w/o transformer | 99744 | 124954 | 52820 | 699215 |
| w/ transformer | **99561** | **124683** | **49903** | **642811** |

