# OpenReview forum: "CORE: Collaborative Optimization with Reinforcement Learning and Evolutionary Algorithm for Floorplanning"
_NeurIPS.cc/2025/Conference — NeurIPS 2025 poster_

### Official Review · Reviewer_y8c9 · 2025-06-28

**Clarity:** 3
**Significance:** 2
**Originality:** 2
**Rating:** 4
**Confidence:** 4

**Summary:**

This paper proposes CORE, an Evolutionary Algorithms (EA) and RL-based floorplanning framework for EDA. B*-Tree and clustering-based diversified evolutionary search are introduced for floorplanning optimization, as well as a PPO-based RL+EA framework. Experimental results show that compared with existing methods, CORE achieves a 12.9% improvement in HPWL.

**Questions:**

1. Could you provide additional experimental evidence demonstrating CORE's superiority over existing EA+RL methods?
2. Could you provide further discussion on the positioning of the B*-tree root? Could you explain why not use methods based on orthogonal constraint graph and shortest path spanning tree?
3. Could you provide additional formal proof or experimental evidence that B*-trees are superior to standard ordered search trees?
4. While App. B addresses scalability concerns, could you provide further discussion on the algorithm's scalability and complexity?
5. Could you provide further comparison on CORE with recent commercial tools, including commercial tools (e.g., Cadence Innovus) or macro-placement methods (e.g., DREAMPlace)?

If the authors address my concerns, I may raise my score accordingly.

**Ethical Concerns:**

["NO or VERY MINOR ethics concerns only"]

**Final Justification:**

The authors have addressed most of my concerns in their rebuttal.

**Limitations:**

While App. B addresses scalability concerns, a more rigorous complexity analysis would strengthen the methodological claims. I think this paper has no potential negative societal impact.

**Paper Formatting Concerns:**

This paper has no major formatting issues.

**Quality:**

2

**Strengths And Weaknesses:**

Strengths:
1. Related work will be open-sourced upon acceptance, providing the community with an effective approach for floorplanning using novel RL+EA-based methods.
2. Combining RL+EA is an established methodology. It's happy to see the authors experimentally validate this approach to address floorplanning challenges in EDA.
3. Experimental results demonstrate effective improvements in HPWL by 12.9%.

Weaknesses:
1. The bottom-left positioning of the B*-tree root (Fig. 1) lacks clear justification. What justifies this over central placement or placement at other positions?
2. The authors provide inadequate coverage of related works, particularly regarding EA+RL approaches such as [1][2] and [3]. Additionally, there is a lack of comparative analysis between the proposed method and classic EA+RL techniques.
* [1] Moriarty D E, Schultz A C, Grefenstette J J. Evolutionary algorithms for reinforcement learning[J]. Journal of Artificial Intelligence Research, 1999, 11: 241-276.
* [2] Khadka S, Majumdar S, Nassar T, et al. Collaborative evolutionary reinforcement learning[C]//International conference on machine learning. PMLR, 2019: 3341-3350.
* [3] Bai H, Cheng R, Jin Y. Evolutionary reinforcement learning: A survey[J]. Intelligent Computing, 2023, 2: 0025.
3. No convergence guarantees for EA-RL collaboration. I noticed the authors mentioned other methods may be "trapped in local optima". How do the proposed CORE avoid local optima?
4. I understand that B*-trees can constrain the search space within approximate bounds, but why are they superior to standard ordered search trees for floorplanning? This requires formal proof or experimental analysis in the context of floorplanning problems.
5. No comparison with commercial tools (e.g., Cadence Innovus) or macro-placement methods (e.g., DREAMPlace).
6. Minor formatting issues:
* typos such as "​​Transfermer" and "Concanated" in Figure 3.
* Figures 2-3 display rendering artifacts ($?_?$) in MS Edge, though they appear correctly in other PDF readers (e.g., Adobe Acrobat). I recommend migrating to cross-platform fonts.

---

> ### Author Rebuttal · Authors · 2025-07-31
>
> We appreciate the reviewer's valuable review and constructive comments, and we would like you to know that your questions provide considerably helpful guidance to improve the quality of our paper.
>
> We will try our best to address each of the concerns and questions raised by the reviewer below:
>
> **1. [Re: The bottom-left positioning of the B\*-tree root (Fig. 1) lacks clear justification. What justifies this over central placement or placement at other positions? Could you provide further discussion on the positioning of the B\*-tree root? ]**
>
> B*-tree is an ordered binary tree, where the root node must be placed first and is defined using a bottom-left positioning scheme. Newly added modules are inserted as left or right children, causing the overall tree to grow toward the right and upward.
>
> We follow the original definition of the B*-tree, as stated in the literature: “A B-tree is an ordered binary tree whose root corresponds to the module on the bottom-left corner.”
>
> [1]. B*-Trees: A New Representation for Non-Slicing Floorplans
>
> **2. [Re: Could you explain why not use methods based on orthogonal constraint graph and shortest path spanning tree?]**
>
> We choose the **B*-tree representation** for three main reasons:
>
> 1. **Efficient evaluation and update.** Since the floorplanner is embedded as the environment of an MDP, each state–action transition must be evaluated quickly. Unlike other representations, the B*-tree does not require the construction of horizontal/vertical constraint graphs or shortest-path computations to evaluate performance, which significantly reduces computational overhead. Moreover, it supports incremental evaluation, making it well-suited for sequential decision-making settings.
> 2. **Consise search space.** A B*-tree encodes fewer permutations compared to most alternative representations, resulting in a more concise search space that facilitates efficient exploration and learning.
> 3. **Modifiable open-source infrastructure.** Effective heuristics such as SA have already been developed for B*-trees, and robust open-source implementations are readily available, making it easier to integrate and extend.
>
> We opt for the B*-tree representation, but our approach is not tied to it. Any other encoding could be substituted, since our work concentrates on the higher-level framework; the core contribution is the development of a cooperative optimization scheme.
>
> **3. [Re: The authors provide inadequate coverage of related works, particularly regarding EA+RL approaches such as [1][2] and [3]. Additionally, there is a lack of comparative analysis between the proposed method and classic EA+RL techniques. [1] Moriarty D E, Schultz A C, Grefenstette J J. Evolutionary algorithms for reinforcement learning[J]. Journal of Artificial Intelligence Research, 1999, 11: 241-276.  Khadka S, Majumdar S, Nassar T, et al. Collaborative evolutionary reinforcement learning[C]//International conference on machine learning. PMLR, 2019: 3341-3350. Bai H, Cheng R, Jin Y. Evolutionary reinforcement learning: A survey[J]. Intelligent Computing, 2023, 2: 0025.]**
>
> We sincerely thank the reviewer for the insightful comment. The algorithms highlighted by the reviewer primarily focus on optimizing policy parameters using EA or RL. In contrast, CORE adopts a direct solution-space optimization approach via EA, which operates in a significantly smaller and more tractable search space. This design choice enhances optimization efficiency compared to methods that rely on parameter-space exploration.
>
> To address the reviewer’s concern, we additionally implemented a parameter-centric ERL framework. It is important to note that such frameworks typically rely on off-policy algorithms, such as TD3 or SAC, which face inherent challenges when applied to discrete action spaces. To ensure a fair and consistent comparison, we adapt the baseline methods to use B* tree-based PPO, which is better suited for discrete structural representations. The corresponding experimental results are presented below:
>
> |  | n10 | n30 | n50 | n100 |
> | --- | --- | --- | --- | --- |
> | ERL | 34645 | 99911 | 124418 | 190343 |
> | CORE | 34296 | 98713 | 123195 | 188376 |
>
> We observe that CORE consistently outperforms the ERL-based methods across all cases. To enhance the completeness of the manuscript, we will include this additional discussion in the revised version. If we have overlooked any relevant work, we would greatly appreciate it if the reviewer could kindly point it out.
>
> **4. [Re: No convergence guarantees for EA-RL collaboration. I noticed the authors mentioned other methods may be "trapped in local optima". How do the proposed CORE avoid local optima?]**
>
> EA excels at global exploration and is capable of rapidly identifying promising solutions, but it often struggles with fine-tuning due to its inherent randomness. In contrast, RL is well-suited for local refinement and policy improvement. To avoid suboptimal solutions and fully leverage the strengths of both paradigms, we design a collaborative mechanism: EA guides RL by providing high-quality samples that help shape its learning direction, while RL in turn injects refined policies into the EA population to steer evolution more effectively. This bidirectional synergy enables efficient cooperation between the two components.
>
> We present the results of our ablation study below, which clearly demonstrate that the integration of both mechanisms leads to improved performance.
>
> |  | n30 | n50 | ami33 | ami49 |
> | --- | --- | --- | --- | --- |
> | CORE w/ Injection | 98888 | 123938 | 49061 | 641670 |
> | CORE w/ IL | 99302 | 124172 | 48403 | 635198 |
> | CORE w/o both | 99561 | 124683 | 49903 | 642811 |
> | CORE | **98713** | **123195** | **46849** | **627620** |
> **5. [Re: why are they superior to standard **ordered search trees** for floorplanning? This requires formal proof or experimental analysis in the context of floorplanning problems.Could you provide additional formal proof or experimental evidence that B\*-trees are superior to **standard ordered search trees**?]**
>
> The original B*-tree paper [1] provides theoretical justifications and empirical results demonstrating several advantages over other tree-based representations:
>
> 1. **Fixed binary structure enables fast operations.** Due to its fixed binary nature, insertion and lookup require only simple pointer manipulations in **O(1) to O(log n)** time, making it significantly lighter than representations like the O-tree, which involve variable branching. This allows for **O(n)** evaluation per layout, making B*-tree well-suited for high-frequency calls in search or reinforcement learning environments.
> 2. **No need for constraint graph construction.** Unlike sequence pair or constraint-graph-based methods, the B*-tree allows direct area computation without constructing horizontal or vertical constraint graphs, thus avoiding shortest-path computations and reducing overhead.
> 3. **Efficient incremental updates.** After module swaps, only the affected nodes along the DFS traversal path require local incremental updates, keeping computational costs low.
>
> These advantages have been empirically validated on MCNC benchmarks, where B*-tree implementations demonstrate faster runtime, lower memory usage, and superior solution quality.
>
> **6. [Re: No comparison with commercial tools (e.g., Cadence Innovus) or macro-placement methods.]**
>
> We thank the reviewer for the suggestion. Cadence Innovus relies on human expert intervention, while DREAMPlace is a macro placement algorithm that typically struggles to produce high-quality results on floorplanning problems. In our work, we provide comparisons with two state-of-the-art macro placement algorithms.
>
> |  | n10 | n30 | n50 | n100 | n200 | n300 | ami33 | ami49 |
> | --- | --- | --- | --- | --- | --- | --- | --- | --- |
> | MaskPlace | 54105 | 150706 | 201743 | 355918 | 666599 | 960487 | 126360 | 2410415 |
> | ChipFormer | 54954 | 122035 | 165600 | 264180 | 500447 | 693501 | 88129 | 1377496 |
> | CORE | 35170 |  99964 | 124890 | 191349 | 348956 | 469242 | 52672 |  656068 |
>
> We can observe that CORE significantly outperforms both MaskPlace and ChipFormer, further demonstrating the effectiveness of methods specifically designed for floorplanning.
>
> **7. [Re: While App. B addresses scalability concerns, could you provide further discussion on the algorithm's scalability and complexity? ]**
>
> CORE introduces RL, which results in additional time overhead compared to heuristic methods. Thus we design a parallel architecture to improve efficiency. Unfortunately, existing learning-based methods for floorplanning have not released open-source implementations. Therefore, we compare the runtime and achieved HPWL with the available baselines.
>
> |  | EA | Maskplace | CORE |
> | --- | --- | --- | --- |
> | n30 | 1.65 \|  104045 | 4.15 \| 150706 | 3.78 \|  99964 |
> | n50 | 3.54 \|  126047 | 6.37 \|  201743 |  8.3 \| 124890 |
>
> CORE achieves better HPWL compared to baselines, and requires more time than EA. However, it outperforms other learning-based method MaskPlace.
>
> Furthermore, CORE can be further accelerated by switching to a statically compiled framework such as JAX. That said, we acknowledge that CORE still has limitations in terms of runtime efficiency. Nevertheless, we believe that learning-based methods represent a promising direction with the potential to deliver competitive solutions.
>
> ---
>
> We hope our replies have addressed the concerns the reviewer posed and shown the improved quality of the paper. **We are always willing to answer any of the reviewer's concerns about our work** and we are looking forward to more inspiring discussions.

---

> ### Comment · Reviewer_y8c9 · 2025-08-02
>
> I am pleased that the authors have addressed most of my concerns in their rebuttal. The additional experiments are impressive, leading me to elevate my overall assessment to a score of 4 (Borderline accept). I maintain openness to score improvement pending further manuscript revisions.

---

> > ### Author Response · Authors · 2025-08-02
> > **Sincerely appreciate your recognition and support!**
> >
> > We are pleased to have addressed the reviewer’s concerns and sincerely appreciate your recognition and support of our work. The constructive suggestions are greatly helpful in improving the quality of our paper.
> >
> > As this year’s *NeurIPS policy does not allow submitting revised manuscripts or providing anonymous links*, we are pleased to outline our planned revisions here:
> >
> > - Provide a more detailed explanation of the B*-tree structure, along with the motivation behind its selection.
> > - Include citations and in-depth discussions of the related works mentioned during the discussion. We would sincerely appreciate it if the reviewers could point out any relevant references we may have missed.
> > - Incorporate the additional experiments into the revised version.
> > - Update the figures and replace the fonts with cross-platform compatible alternatives.
> > - Correct the typos identified by the reviewer.
> >
> > Each of the aforementioned details will be carefully reviewed and addressed with precision in our revised submission.
> > **If the reviewers have any remaining concerns that have not been addressed, or if there are any other suggestions to further improve the paper, please feel free to inform us at any time.**
> >
> > Thank you again for your thoughtful feedback and the valuable discussions.

---

### Official Review · Reviewer_ayg2 · 2025-07-02

**Clarity:** 2
**Significance:** 2
**Originality:** 2
**Rating:** 4
**Confidence:** 4

**Summary:**

In this paper the authors propose an approach combining reinforcement learning and evolutionary algorithms for floor planning in chip design, all in a B*-Tree representation. They describe the approach in terms of the two parts (RL and EA) separately and then how they can influence one another. They then present their results compared to several other floor planning baselines. They are able to consistently outperform these baselines in terms of wire length. The authors also include an ablation and some additional analysis, which supports the importance of both the EA and RL components of their approach.

**Questions:**

1. Can the authors provide some clarity on their EA approach related to my concerns above?
2. Is there a reason the authors don't mention MAP-Elites in the paper?

**Ethical Concerns:**

["NO or VERY MINOR ethics concerns only"]

**Final Justification:**

Based on the discussion and the other reviews I have chosen to maintain my score. I acknowledge the value of the approach but I'm not sure it is sufficient for me to move beyond a borderline accept.

**Limitations:**

yes

**Quality:**

4

**Strengths And Weaknesses:**

# Quality

This represents a very well conducted research project. I appreciate the concept of combining RL and EAs for floorplanning. The evaluation is well-argued and supports the authors' claims and the ablation and additional graphs support the authors' choices behind their approach. I have no major concerns with the quality of the work. I do have some concerns with the quality of the paper which I'll get to next.

# Clarity

I think this paper could be improved substantially. At present, the authors do not include their algorithm pseudocode in the main text of the paper. This is very confusing and essentially means that readers must read the appendices to understand their approach, which is against the formatting guide.

On a more minor note, the authors repeatedly refer to their approach as an evolutionary algorithm, but it's unclear if it has a crossover function. The concept of selecting parents is mentioned once, but it seems this may only have been meant in terms of which members of a population are mutated. Similarly I'm unclear why EAs are referred to as black box optimization methods when their process is transparent. Clarity around these points would be appreciated. I have another more substantial concern around the EA but I'll leave that for the originality section below.

# Significance

This work would be fairly significant within the Electronic Design Automation community. It would also be significant to some extent within the broader AI community due to the clever combination of EAs and RL. But I think this may be the weakest attribute of the paper for this venue, since the contributions are not really to neural information processing systems.

# Originality

This is not the first work to combine RL and EAs but it may be the first work to combine these approaches to floorplanning and is certainly the first work to combine these with B*-Trees. That would lead to a fairly minor originality all considered. However, I have one major concern which is that the authors appear to have potentially reinvented MAP-Elites. The authors make use of quality and diversity search based on binning solutions, which is very close to MAP-Elites. Despite this, the authors do not mention MAP-Elites or even directly identify their work as falling within the Quality-Diversity (QD) paradigm. If the authors are unfamiliar with this area of EA research I'd recommend looking into it.

---

> ### Author Rebuttal · Authors · 2025-07-31
>
> We appreciate the reviewer's valuable review and constructive comments, and we would like you to know that your questions provide considerably helpful guidance to improve the quality of our paper.
>
> We will try our best to address each of the concerns and questions raised by the reviewer below:
>
> **1. [Re: The authors do not include their algorithm pseudocode in the main text of the paper.]**
>
> We thank the reviewer for the valuable suggestion. We will incorporate the pseudocode into the main text to improve readability in the revised version.
>
> **2. [Re: the authors repeatedly refer to their approach as an evolutionary algorithm, but it's unclear if it has a crossover function. The concept of selecting parents is mentioned once, but it seems this may only have been meant in terms of which members of a population are mutated.]**
>
> We thank the reviewer for the valuable suggestion. We would like to clarify that our algorithm does not employ crossover operations. The primary reason is that crossover is difficult to apply to tree-based representations. For instance, in two parent trees, each node corresponds to a different module. Directly exchanging subtrees may lead to module duplication.
>
> **3. [Re: I'm unclear why EAs are referred to as black box optimization methods when their process is transparent. Clarity around these points would be appreciated.]**
>
> A black-box optimization algorithm refers to an approach that only requires access to the objective function (fitness function), without needing any explicit gradients, constraint formulations, or internal structural information. EAs fall under this category: although their optimization process is transparent, they rely solely on the evaluation of solution scores f(x), without analytically exploiting the structure of f(x) to guide the optimization of x. For this reason, EAs are commonly referred to as black-box optimization methods.
>
> **4. [Re: I have one major concern which is that the authors appear to have potentially reinvented MAP-Elites. The authors make use of quality and diversity search based on binning solutions, which is very close to MAP-Elites. Despite this, the authors do not mention MAP-Elites or even directly identify their work as falling within the Quality-Diversity (QD) paradigm.]**
>
> We thank the reviewer for pointing out this omission. MAP-Elites typically requires the explicit construction of an archive to store the best-performing solutions across different behavioral descriptors. In contrast, our method employs clustering to identify diverse and high-quality solutions, thereby mitigating the risk of convergence to suboptimal regions.
>
> Our results confirm that incorporating quality diversity leads to more efficient optimization and yields higher-quality solutions compared to a standard EA baseline. (See the table below for details.)
>
> |  | n30 | n50 | ami33 | ami49 |
> | --- | --- | --- | --- | --- |
> | CORE w/ EA | 101252 | 127929 | 52604 | 698285 |
> | CORE w/ CDES | 98713 | 123195 | 46849 | 627620 |
>
> Thank you for the valuable suggestions. We will include this discussion about MAP-Elites in the revised version. We would greatly appreciate it if the reviewer could point out any relevant references we may have overlooked.
>
>
> ---
>
> We hope our replies have addressed the concerns the reviewer posed and shown the improved quality of the paper. **We are always willing to answer any of the reviewer's concerns about our work** and we are looking forward to more inspiring discussions.

---

> > ### Comment · Reviewer_ayg2 · 2025-08-04
> > **Re: Rebuttal by Authors**
> >
> > Thanks to the authors for their rebuttal. I am satisfied by the authors' response to (1) and (3).
> >
> > For (2), can I get some clarity in terms of how the authors are defining evolutionary algorithms then? Crossover is typically considered a standard element of such an algorithm from the literature I'm familiar with.
> >
> > For (4), can the authors clarify what this standard EA baseline was? Relatedly, are the authors saying that their algorithm now does fall within a quality-diversity paradigm?

---

> > > ### Author Response · Authors · 2025-08-04
> > > **We sincerely appreciate the reviewer’s prompt and constructive feedback.**
> > >
> > > We sincerely appreciate the reviewer’s prompt and constructive response. We are pleased to have addressed some of the concerns raised, and below we provide our responses to the additional questions.
> > >
> > > 1. **[Re: can I get some clarity in terms of how the authors are defining evolutionary algorithms then? Crossover is typically considered a standard element of such an algorithm from the literature I'm familiar with.]**
> > >
> > > The generalized definition of evolutionary algorithms should encompass the following workflow: **individual/genome representation**, **population initialization**, **fitness evaluation**, **parent selection**, and **variation operators**. This framework serves to unify various branches such as Genetic Algorithms, Evolution Strategies, Evolutionary Programming, and Genetic Programming.
> > >
> > > Among them, **variation operators may include crossover operators, mutation operators, as well as customized operators.**
> > > But crossover operators are not strictly necessary in evolutionary algorithms. Although commonly used in traditional Genetic Algorithms and quality diversity approaches like MAP-Elites, crossover is generally considered an *enhancement operator* rather than a fundamental component.
> > >
> > > A representative example is the (1+1) EA, which maintains only a single individual as the current solution. It generates new candidates through mutation alone, and updates the current solution based on fitness comparison, without involving crossover operator.
> > >
> > > Moreover, the Estimation of Distribution Algorithm (EDA) is a major branch of evolutionary algorithms that does not include crossover. Instead, it builds a probabilistic model to characterize the distribution of promising solutions and samples new candidates from this model, updating it based on the superior sampled individuals.
> > >
> > > In addition, some nature-inspired swarm intelligence algorithms are also commonly regarded as part of evolutionary computation. These methods likewise do not explicitly define crossover operators, such as Particle Swarm Optimization and Ant Colony Optimization,.
> > >
> > > In summary, crossover is one type of variation operator in EAs. While useful in some contexts, it is not essential for the design or functioning of an evolutionary algorithm.
> > >
> > > If the reviewer holds a different view, we are open to further discussion and would greatly value your insights.
> > >
> > > 2. **[Re: can the authors clarify what this standard EA baseline was? ]**
> > >
> > > The standard EA is implemented based on the B*-tree. The only difference from CDES lies in the removal of the clustering-based diversity maintenance mechanism. Instead of maintaining multiple clusters, it evolves directly over the entire population, using exactly the same operators. We believe this comparison is fair and can clearly demonstrate the effectiveness of the diversity mechanism.
> > >
> > > 3. **[Re: Relatedly, are the authors saying that their algorithm now does fall within a quality-diversity paradigm?]**
> > >
> > > Yes, we do consider our method to fall into the QD paradigm, as this mechanism plays a crucial role in avoiding sub-optimal solutions. We sincerely thank the reviewer for the insightful suggestion and reminder. We will include a dedicated subsection on QD in the *Related Work* section, where we will elaborate on its relevance to our approach. This will include discussions of representative works such as MAP-Elites, Novelty Search, CMA-ME, ME-ES, PGA-ME, MEGA, and AURORA. **If there are any additional works we may have overlooked, we would be truly grateful if the reviewer could kindly point them out.**
> > >
> > > We sincerely thank the reviewer for the thoughtful feedback and constructive discussion. **If there are any remaining concerns or further suggestions for improvement, please do not hesitate to let us know, we are fully committed to addressing them to the best of our ability**. Once again, we greatly appreciate your valuable time and insightful comments.

---

> > > > ### Comment · Reviewer_ayg2 · 2025-08-05
> > > > **Re: We sincerely appreciate the reviewer’s prompt and constructive feedback.**
> > > >
> > > > Thanks to the authors for their answers, that clarifies their perspective. Reading the other reviews and discussion I have decided to maintain my score.

---

> > > > > ### Author Response · Authors · 2025-08-06
> > > > > **Deeply appreciate the valuable suggestions and discussions!**
> > > > >
> > > > > We sincerely thank you for the valuable and in-depth discussions, as well as for your recognition and support of our work. Your constructive feedback and encouragement have been truly motivating for us.
> > > > >
> > > > > We will ensure that all necessary clarifications and improvements are thoroughly and appropriately incorporated into the revised version to benefit the floorplanning community.
> > > > >
> > > > > **Thank you again for your time, thoughtful engagement, and kind support throughout the review process.**

---

### Official Review · Reviewer_F7Fd · 2025-07-02

**Clarity:** 3
**Significance:** 3
**Originality:** 3
**Rating:** 4
**Confidence:** 3

**Summary:**

This paper proposes CORE (Collaborative Optimization with RL and Evolutionary Algorithm), a hybrid framework designed to address the floorplanning problem by improving search and exploration to avoid local optima. The central idea is to synergize the global search capabilities of Evolutionary Algorithms (EAs) with the local, learning-based refinement of Reinforcement Learning (RL). The paper's main contributions are:
1. A novel collaborative framework where the EA and RL components mutually guide each other.
2. An EA component, named Clustering-based Diversified Evolutionary Search (CDES), which uses layout feature clustering and a novelty-based fitness score to maintain a diverse population of solutions, enhancing global exploration.
3. An RL component models the floorplanning task as the sequential construction of a B*-Tree, a compact and non-slicing floorplan representation.
4. Two specific collaboration mechanisms: a "Reinforcement-Driven" mechanism where high-quality layouts from the RL agent are injected into the EA's population, and an "Evolutionary-Guided" mechanism where superior layouts found by the EA are used as expert demonstrations to guide the RL agent via imitation learning.

Experimental results on the MCNC and GSRC benchmarks show that CORE outperforms strong baselines in wirelength (HPWL) and area utilization.

**Questions:**

1. You run each algorithm five times and reporting the best convergence performance achieved. I am not sure if it is a convention in this area. Why not report the mean and standard deviation?
2. Your work shows excellent results on HPWL and AU optimization. A key challenge in industrial floorplanning is satisfying a variety of specific hardware design rules, such as boundary alignment or grouping constraints. How might the CORE framework, particularly its B*-Tree representation, be adapted to handle such constraints? Would this require new mutation operators for the EA and a modified reward/MDP formulation for the RL component?
3. The "Evolutionary-Guidance" mechanism relies on converting a complete B*-Tree from the EA archive into a state-action sequence for the RL agent to imitate. Could you elaborate on this conversion process? Since the RL agent builds the tree sequentially, the order of node insertions matters. Is this order uniquely determined from a static B*-Tree, or does it require a specific traversal (e.g., pre-order)? How does this choice of conversion affect the stability and effectiveness of the imitation learning?
4. The analysis in Figure 6 suggests a healthy, bidirectional collaboration. Do the roles of the EA and RL shift over the course of training? For example, one might hypothesize that the EA's guidance is more critical in the early stages to help the RL agent find promising regions of the search space, while the RL agent's injections are more impactful later for fine-tuning the population. Do your observations support this, or is the collaborative balance relatively stable?
5. The use of B*-Tree is a key design choice. Did you consider other compact representations like Sequence-Pair or Corner Block Lists? What are the specific advantages of B*-Tree for this hybrid EA+RL framework that made it the preferred choice?

**Ethical Concerns:**

["NO or VERY MINOR ethics concerns only"]

**Final Justification:**

The rebuttal addressed most of my concerns.

**Limitations:**

The paper mentions that "it discuss the limitations in Appendix B" (Line 647). However, Appendix B describes a parallelization framework to reduce runtime.

**Quality:**

3

**Strengths And Weaknesses:**

# Strengths
1. The hybrid EA+RL approach is a well-motivated and powerful idea for complex optimization problems like floorplanning. Combining the global exploration of an EA with the local learning of RL is an effective paradigm. The specific two-way collaboration mechanisms (RL-to-EA injection and EA-to-RL imitation learning) are well-designed and novel in this context.
2. The RL formulation is based on sequentially constructing a B*-Tree. This is a clever and powerful approach, as the B*-Tree representation inherently guarantees compact, non-overlapping layouts, fundamentally shifting the RL problem from direct coordinate placement to making structured decisions within a combinatorial tree space.
3. The paper is backed by a strong empirical study. The ablation study in Table 2 convincingly demonstrates that the full CORE framework outperforms both the standalone EA (CDES) and RL (PPO) components, providing clear evidence of synergy. Furthermore, the analysis in Figure 6 provides excellent insight into the collaborative dynamics, showing that both guidance mechanisms remain active and effective throughout training. The performance gains in HPWL over strong baselines are substantial.
4. The paper is clearly written and well-organized. The high-level framework is effectively communicated through diagrams (Fig. 2 and Fig. 4), which are essential for understanding the complex interactions between the different components.

# Weaknesses
1. The paper's evaluation focuses exclusively on optimizing foundational metrics like HPWL and Area Utilization. It does not address the broader set of specific, complex hardware design rules (e.g., boundary constraints, grouping constraints) that are often critical in modern industrial floorplanning. This narrows the immediate applicability of the proposed method to more complex, real-world design scenarios.
2. For two RL baselines, the Area Utilization was an assumption taken from the literature, not a direct measurement, which slightly weakens the comparison on that metric.
3. The CORE framework is significantly complex, integrating a diversity-driven EA, a GNN/Transformer-based PPO agent, and multiple collaboration mechanisms. This results in a large number of hyperparameters (Table 4) and moving parts, which could pose a challenge for reproduction and practical implementation.

---

> ### Author Rebuttal · Authors · 2025-07-31
>
> We appreciate the reviewer's valuable review and constructive comments, and we would like you to know that your questions provide considerably helpful guidance to improve the quality of our paper.
>
> We will try our best to address each of the concerns and questions raised by the reviewer below:
>
> **1. [Re: For two RL baselines, the Area Utilization was an assumption taken from the literature, not a direct measurement, which slightly weakens the comparison on that metric. ]**
>
> Consequently, for several metrics that were not reported in the original papers, we provide results under idealized assumptions. To mitigate these issues and ensure reproducibility and fair benchmarking, we will release our algorithm along with a Python-based floorplanning platform.
>
> **2. [Re: You run each algorithm five times and reporting the best convergence performance achieved. I am not sure if it is a convention in this area. Why not report the mean and standard deviation?]**
>
> We follow the prior works by reporting the best results, as this highlights the algorithm’s peak performance. For completeness, we also provide the mean and standard deviation in the appendix, as shown in the table below.
>
> |  | n10 | n30 | n50 | n100 | n200 | n300 | ami33 |
> | --- | --- | --- | --- | --- | --- | --- | --- |
> | Best baseline | 36044 | 108835 | 137516 | 218976 | 351735 | 476765 | 64315 |
> | CORE | 35290 ± 294 | 100733 ± 1113 | 125132 ± 473 | 191610 ± 840 | 348416 ± 1140 | 470296 ± 701 | 52930 ± 753 |
>
> We can observe that CORE still consistently outperforms the best performance of other baselines.
>
> **3. [Re: A key challenge in industrial floorplanning is satisfying a variety of specific hardware design rules, such as boundary alignment or grouping constraints. How might the CORE framework, particularly its B\*-Tree representation, be adapted to handle such constraints? Would this require new mutation operators for the EA and a modified reward/MDP formulation for the RL component?]**
>
> Adding constraints typically does not require changes to the algorithm itself. Instead, we can incorporate the constraints into the reward or fitness function. For example, by embedding grouping constraints as part of the evaluation objective—both in the fitness function for EA and the reward function for RL. we can handle such constraints within the existing optimization framework. In addition, some adjustments based on the specific optimization objectives may also be necessary. For example, constructing subtrees on top of the B* tree can be an effective way to address grouping constraints.
>
> **4. [Re: converting a complete B\*-Tree from the EA archive into a state-action sequence for the RL agent to imitate.  or does it require a specific traversal (e.g., pre-order)? How does this choice of conversion affect the stability and effectiveness of the imitation learning? ]**
>
> Before answering this question, it is important to outline the optimization process of the B*-tree. After each module is inserted, the B*-tree adjusts the entire layout to ensure seamless placement of modules. This adjustment is performed in a depth-first traversal of the tree, during which each node is updated sequentially, and the adjustment order is recorded.
>
> Consequently, when converting a layout back into a decision sequence, we must model it according to this recorded order. The reason is that any deviation from the original traversal order can lead to inconsistencies between the reconstructed and original layouts. For example, adjusting the left or right child first may result in entirely different outcomes. This process is inherently tied to the structure of the B*-tree, which relies on depth-first traversal for both module placement and reordering. Therefore, reconstructing the sequence must also follow the same depth-first traversal logic.
>
> **5. [Re: Do the roles of the EA and RL shift over the course of training? For example, one might hypothesize that the EA's guidance is more critical in the early stages to help the RL agent find promising regions of the search space, while the RL agent's injections are more impactful later for fine-tuning the population. Do your observations support this, or is the collaborative balance relatively stable?]**
>
> We observe a three-phase phenomenon: (1) **Initial Phase**: RL) dominates. (2) **Middle Phase**: The effect of EA gradually becomes more prominent, while the impact of RL diminishes. (3) **Final Phase**: RL regains influence and starts contributing effectively again.
>
> The reason behind Phase 1 is that, in our experiments, layouts generated by randomly initialized policy networks often outperform those constructed by random layout initialization by EA. As a result, the RL injection success rate reaches nearly 100% in the early stages.
>
> As EA proceeds through several iterations, it begins to discover solutions that outperform those generated by RL, leading to a drop in the RL injection success rate.
>
> After a period of training, we observe that the quality of EA-generated solutions relative to RL begins to decline, and the RL injection success rate starts to rise again.
>
> Except for the first phase, our findings are consistent with the reviewer’s observations in the second and third phases. The first phase is a notable exception, which we attribute to the fundamental difference between the two paradigms: EA focuses on solution optimization, whereas RL optimizes the policy. This structural difference leads to a performance gap in initial solution quality.
>
> **6. [Re: Did you consider other compact representations like Sequence-Pair or Corner Block Lists? What are the specific advantages of B\*-Tree for this hybrid EA+RL framework that made it the preferred choice?]**
>
> We choose the **B*-tree representation** for three main reasons:
>
> 1. **Efficient evaluation and update.** Since the floorplanner is embedded as the environment of an MDP, each state–action transition must be evaluated quickly. Unlike other representations, the B*-tree does not require the construction of horizontal/vertical constraint graphs or shortest-path computations to evaluate performance, which significantly reduces computational overhead. Moreover, it supports incremental evaluation, making it well-suited for sequential decision-making settings.
> 2. **Consise search space.** A B*-tree encodes fewer permutations compared to most alternative representations, resulting in a more concise search space that facilitates efficient exploration and learning.
> 3. **Modifiable open-source infrastructure.** Effective heuristics such as SA have already been developed for B*-trees, and robust open-source implementations are readily available, making it easier to integrate and extend.
>
> We opt for the B*-tree representation, but our approach is not tied to it. Any other encoding could be substituted, since our work concentrates on the higher-level framework; the core contribution is the development of a cooperative optimization scheme.
>
> **7. **[Re: Limitations]****
>
> We thank the reviewer for the valuable suggestion. Below, we provide a discussion of some limitations of CORE.
>
> 1. **Time overhead**: Learning-based methods typically incur additional training costs. Although we mitigate this through a parallel framework or potential framework modifications to reduce training time, the process remains relatively time-consuming.
> 2. **Theoretical support**: Our method currently lacks theoretical guarantees and relies more on empirical validation.
>
> ---
>
> We hope our replies have addressed the concerns the reviewer posed and shown the improved quality of the paper. **We are always willing to answer any of the reviewer's concerns about our work** and we are looking forward to more inspiring discussions.

---

> > ### Author Response · Authors · 2025-08-02
> > **Additional Clarifications and Supporting Experiments**
> >
> > We sincerely apologize, we just realized that our previous response to the first question included an incomplete answer.
> > We would like to provide the following additional clarification:
> >
> > 1. **[Re: For two RL baselines, the Area Utilization was an assumption taken from the literature, not a direct measurement.]**
> >
> > For some methods, we are unable to find publicly available source code or executable binaries. In fact, most RL-based floorplanning methods are typically not open-sourced. Consequently, for several metrics that were not reported in the original papers, we provide results under idealized assumptions. To mitigate these issues and ensure reproducibility, **we will release our algorithm along with a Python-based floorplanning platform.**
> >
> > 2. **[Re: Hyperparameter Selection and Analysis]**
> >
> > CORE can be conceptually divided into three components: EA design, RL design, and EA-RL collaboration.
> >
> > - **On the EA side**，we first analyze CDES independently (excluding the influence of RL), with the objective of minimizing HPWL. The results are as follows:
> >
> > | Pop size | n50 | n100 | ami49 |
> > | --- | --- | --- | --- |
> > | 50 | 126147 | 211774 | 711931 |
> > | 100 | 126047 | 199488 | **679473** |
> > | 200 | **124849** | **198223** | 680698 |
> >
> > We observe that larger population sizes generally lead to better performance.
> >
> > | Cluster | n50 | n100 | n200 | ami49 |
> > | --- | --- | --- | --- | --- |
> > | 1 | 132708 | 229737 | 417632 | 790673 |
> > | 2 | 127841 | 226644 | 403261 | 759635 |
> > | 4 | **126047** | 199488 | 382955 | **679473** |
> > | 10 | 144938 | **192398** | **357297** | 993230 |
> >
> > | Elite size e | n50 | n100 | n200 | ami49 |
> > | --- | --- | --- | --- | --- |
> > | 1 | 135760 | **191708** | **359261** | 824228 |
> > | 2 | **126047** | 199488 | 382955 | **679473** |
> > | 5 | 130018 | 225533 | 410475 | 731270 |
> >
> > For the number of clusters, a setting of 4 performs better on smaller-scale problems, while 10 is more effective for larger-scale instances.
> >
> > A similar trend is seen with the number of elites: n = 1 performs better on larger-scale tasks, whereas n=2 works better for smaller-scale cases.
> >
> > | α | n50 | n100 | n200 | ami49 |
> > | --- | --- | --- | --- | --- |
> > | 0.0 | 138954 | 234500 | 408446 | 837418 |
> > | 0.2 | 138592 | 231034 | 411872 | 805013 |
> > | 0.5 | 136496 | 225504 | 418917 | 766239 |
> > | 0.8 | **126047** | **199488** | **382955** | **679473** |
> > | 1.0 | 191069 | 341052 | 835619 | 1729777 |
> >
> > For α,  0.8 yields better performance, achieving a more balanced trade-off between novelty and quality.
> >
> > **It is worth noting that we did not fine-tune these parameters—across all tasks, we consistently used 4 clusters, elite size 2, α = 0.8.**
> >
> > - **On the RL side**, we first conducted an ablation study on the policy architecture.
> >
> > It is important to note that the network input includes a netlist, whose topological structure naturally lends itself to graph neural networks (GNNs). Therefore, the use of a GNN is essential. We introduce transformer layers primarily for better representation. The ablation study are shown below:
> >
> > |  | n30 | n50 | ami33 | ami49 |
> > | --- | --- | --- | --- | --- |
> > | w/o transformer | 99744 | 124954 | 52820 | 699215 |
> > | w/ transformer | **99561** | **124683** | **49903** | **642811** |
> >
> > - **On the collaboration side**, we conduct following experiments.
> >
> > |  | n30 | n50 | ami33 | ami49 |
> > | --- | --- | --- | --- | --- |
> > | CORE w/ Injection | 98888 | 123938 | 49061 | 641670 |
> > | CORE w/ IL | 99302 | 124172 | 48403 | 635198 |
> > | CORE w/o both | 99561 | 124683 | 49903 | 642811 |
> > | CORE | **98713** | **123195** | **46849** | **627620** |
> >
> > We observe that the best performance is achieved only when both mechanisms are used together.
> >
> > Next, we analyze the imitation coefficient w_{IL}.
> >
> > | w_{IL} | n50 | n100 | ami33 | ami49 |
> > | --- | --- | --- | --- | --- |
> > | 1.0 | 124319 | 189956 | 48437 | 634400 |
> > | 0.1 | **123195** | **188376** |  **46849** | **627620** |
> > | 0.01 | 124054 | 189103 | 47619 | 631599 |
> > | 0.0 | 124172 | 189009 | 48403 | 635198 |
> >
> > We observe that the algorithm performs best when the coefficient is set to 0.1.
> >
> > Overall, we explored several algorithmic design choices with the goal of better addressing the NP-hard nature of floorplanning. For all subsequent experiments, **we kept the hyperparameters fixed across cases**. While we acknowledge that these hyperparameters may not be optimal, we found them sufficient to achieve satisfactory results across all test cases.
> >
> > **To ensure reproducibility, we will open-source our algorithm, which we believe is crucial for progress in the learning-based floorplanning.**
> >
> > We sincerely welcome further discussion with the reviewer. If our responses have helped to clarify the concerns, *we would be grateful if the reviewer would consider re-evaluating our work*. **If any questions or uncertainties remain, please do not hesitate to let us know — we will make every effort to address them thoroughly**. Thank you again for your time and thoughtful feedback.

---

### Official Review · Reviewer_YTf4 · 2025-07-03

**Clarity:** 3
**Significance:** 3
**Originality:** 3
**Rating:** 4
**Confidence:** 3

**Summary:**

This work addresses the floorplanning problem and proposes a hybrid framework aimed at achieving efficient and high-quality solutions. The framework integrates Evolutionary Algorithms (EAs) with Reinforcement Learning (RL) in a synergistic manner. Specifically, the authors introduce a Clustering-based Diversified Evolutionary Search, formulate floorplanning as a sequential decision-making problem using a B*-Tree representation, and employ RL for efficient policy learning. The coordination between EA and RL is designed to combine their respective strengths. Empirical results demonstrate the effectiveness of the proposed approach.

**Questions:**

How does the runtime efficiency of the proposed method compare to other baseline approaches?

**Ethical Concerns:**

["NO or VERY MINOR ethics concerns only"]

**Limitations:**

A deeper ablation analysis is needed to fully understand the contribution and sensitivity of each component in the proposed framework.

**Quality:**

3

**Strengths And Weaknesses:**

Strengths:

1. Clear and accessible presentation: The paper is well-organized and easy to follow.

2. Sound and novel methodology: The combination of EA and RL is innovative and well-motivated. The use of B*-Tree as a representation mechanism is a clever choice for the floorplanning domain. Both the design of individual components and their integration are thoughtfully executed.

Weaknesses:

1. Potential efficiency concerns: The integration of multiple complex components (EA, RL, B*-Tree) may introduce runtime overhead, which could affect scalability or deployment in real-world settings.

2. Lack of ablation studies: The paper would benefit from more extensive ablation experiments to isolate and validate the contribution of each key component in the framework.

---

> ### Author Rebuttal · Authors · 2025-07-31
>
> We appreciate the reviewer's valuable review and constructive comments, and we would like you to know that your questions provide considerably helpful guidance to improve the quality of our paper.
>
> We will try our best to address each of the concerns and questions raised by the reviewer below:
>
> **1. [Re: The integration of multiple complex components (EA, RL, B\*-Tree) may introduce runtime overhead, which could affect scalability or deployment in real-world settings.]**
>
> Learning-based methods typically need additional runtime overhead, primarily due to the cost of sampling and training. To address this issue, we design a parallel architecture that significantly reduces the runtime burden (details in Appendix B).
>
> Moreover, the framework can be further accelerated by adopting the statically compiled JAX backend instead of PyTorch, which often yields a speedup of approximately 300%.
>
> In practical industrial floorplanning tasks, the number of modules is typically below 50, and cases exceeding this scale are extremely rare. This makes learning-based approaches particularly competitive.
>
> Furthermore, compared to the current state-of-the-art macro placement algorithm, our method achieves comparable runtime while delivering superior performance.
>
> |  | EA | Maskplace | CORE |
> | --- | --- | --- | --- |
> | n30 | 1.65 \|  104045 | 4.15 \| 150706 | 3.78 \|  99964 |
> | n50 | 3.54 \|  126047 | 6.37 \|  201743 | 8.3 \| 124890 |
>
> **2. [Re: More extensive ablation experiments to isolate and validate the contribution of each key component in the framework.]**
>
> We thank the reviewers for their valuable suggestions. In response, we provide the following ablations:
>
>  **Coordination Mechanism:** The ablation results are presented below. As shown, integrating both of our proposed coordination mechanisms leads to the best overall performance.
>
> |  | n30 | n50 | ami33 | ami49 |
> | --- | --- | --- | --- | --- |
> | CORE w/ Injection | 98888 | 123938 | 49061 | 641670 |
> | CORE w/ IL | 99302 | 124172 | 48403 | 635198 |
> | CORE w/o both | 99561 | 124683 | 49903 | 642811 |
> | CORE | **98713** | **123195** | **46849** | **627620** |
>
>
>  **CDES:** We compare our proposed CDES with a vanilla EA that does not incorporate the clustering method. The results show that CDES achieves significantly better optimization performance than the vanilla EA.
>
>  |  | n30 | n50 | ami33 | ami49 |
>  | --- | --- | --- | --- | --- |
>  | CORE w/ EA | 101252 | 127929 | 52604 | 698285 |
> | CORE w/ CDES | 98713 | 123195 | 46849 | 627620 |
> **B\*-Tree Modeling:** We evaluate the effectiveness of the B*-tree modeling approach against alternative structure Sequence pair. The results indicate that the B*-tree model consistently yields superior performance.
>
> |  x 1e3 | n100 | n200 | n300 | ami33 |
> | --- | --- | --- | --- | --- |
> | Sequence pair | 209.8 | 404.5 | 556.9 | 59.0 |
> | B* tree | 191.3 | 349.0 | 369.2 | 52.7 |
>
> **3. [Re: How does the runtime efficiency of the proposed method compare to other baseline approaches?]**
> CORE introduces RL, which results in additional time overhead compared to heuristic methods. Thus we design a parallel architecture to improve efficiency. We compare the runtime and achieved HPWL with available baselines:
> |  | EA | Maskplace | CORE |
> | --- | --- | --- | --- |
> | n30 | 1.65 \|  104045 | 4.15 \| 150706 | 3.78 \|  99964 |
> | n50 | 3.54 \|  126047 | 6.37 \|  201743 | 8.3 \| 124890 |
>
> CORE achieves better HPWL compared to baselines, and requires more time than EA. However, it outperforms other learning-based method MaskPlace.
>
>
> ---
>
> We hope our replies have addressed the concerns the reviewer posed and shown the improved quality of the paper. **We are always willing to answer any of the reviewer's concerns about our work** and we are looking forward to more inspiring discussions.

---

> > ### Comment · Reviewer_YTf4 · 2025-08-06
> >
> > Thank you for your detailed response. I still believe that this paper is a solid paper and will maintain my positive score.

---

> > > ### Author Response · Authors · 2025-08-07
> > > **Sincerely appreciate your recognition and support!**
> > >
> > > We sincerely appreciate your recognition and support of our work. The constructive suggestions are greatly helpful in improving the quality of our paper.
> > >
> > > We will carefully incorporate all manuscript refinements and experiments discussed into the revised version.
> > >
> > > **Thank you again for your thoughtful feedback and the valuable discussions.**

---

### Decision · Program_Chairs · 2025-09-17

**Decision:**

Accept (poster)

**Comment:**

The paper introduces a method for floorplanning which combines evolutionary strategies and reinforcement learning over B* trees.
The method shows significant gains in wirelength with respect to other methods in the literature and the paper is well written.
While the method has a somewhat limited novelty, introduces several hyperparameters and has longer runtime, the authors in the rebuttal have addressed concerned about the flexibility and ablations, providing further empirical evidence.